# Daytime temperature is sensed by phytochrome B in *Arabidopsis* through a transcriptional activator HEMERA

Yongjian Qiu[1], Meina Li[1,2], Ruth Jean-Ae Kim[1], Carisha M. Moore[1] & Meng Chen [ID] [1]

Ambient temperature sensing by phytochrome B (PHYB) in *Arabidopsis* is thought to operate mainly at night. Here we show that PHYB plays an equally critical role in temperature sensing during the daytime. In daytime thermosensing, PHYB signals primarily through the temperature-responsive transcriptional regulator PIF4, which requires the transcriptional activator HEMERA (HMR). HMR does not regulate *PIF4* transcription, instead, it interacts directly with PIF4, to activate the thermoresponsive growth-relevant genes and promote warm-temperature-dependent PIF4 accumulation. A missense allele *hmr-22*, which carries a loss-of-function D516N mutation in HMR's transcriptional activation domain, fails to induce the thermoresponsive genes and PIF4 accumulation. Both defects of *hmr-22* could be rescued by expressing a HMR22 mutant protein fused with the transcriptional activation domain of VP16, suggesting a causal relationship between HMR-mediated activation of PIF4 target-genes and PIF4 accumulation. Together, this study reveals a daytime PHYB-mediated thermosensing mechanism, in which HMR acts as a necessary activator for PIF4-dependent induction of temperature-responsive genes and PIF4 accumulation.

[1] Department of Botany and Plant Sciences, Institute for Integrative Genome Biology, University of California, Riverside, CA 92521, USA. [2] Present address: School of Life Sciences, Guangzhou University, Guangzhou 510006, China. Correspondence and requests for materials should be addressed to M.C. (email: meng.chen@ucr.edu)

Ambient growth temperature profoundly influences almost every facet of plant development and growth[1]. Increases in global temperature have already had dramatic impacts on plant phenology, distribution, diversity, and are expected to significantly decrease crop productivity[2–6]. A molecular understanding of how plants sense and respond to temperature becomes critical to predict the ecological impact of global temperature increases and develop new technologies to cope with climate change.

In *Arabidopsis*, changes in ambient temperature between 12 °C and 27 °C trigger diverse developmental, physiological, and morphological responses, including modulations in shoot and root growth, stomatal differentiation, flowering, immunity, and yield, which are collectively called thermomorphogenesis[1,7–13]. Among these, a widely used readout for thermosensing is the elongation response to warm temperatures of the embryonic stem or hypocotyl[14]. Hypocotyl elongation is controlled by the circadian clock and partitioned to a certain time of the day, which varies between short-day (SD) and long-day (LD) conditions[15–17]. In SD conditions, hypocotyl elongation mainly occurs at the end of night or in the dark[16]. By contrast, in LD, including continuous light conditions, hypocotyl elongation peaks during daytime in the light[15–17]. In accordance with the contrast in daylength-dependent rhythmic growth patterns, hypocotyl elongation is modulated by temperature changes at the distinct times in either the dark or light under SD and LD conditions, respectively[18].

Ambient temperature is sensed by the red (R) and far-red (FR) photoreceptor, phytochrome B (PHYB)[19,20]. PHYB can be photoconverted between two relatively stable forms: the R light-absorbing inactive Pr form and the FR light-absorbing active Pfr form[21,22]. The active PHYB controls almost all aspects of plant development and growth, including restricting hypocotyl elongation[23,24]. The photoconversion by R and FR light enables phytochromes to sense changes in the ratio between R and FR light, which inform the presence of competing neighboring plants as well as the diurnal and seasonal time[25–27], and thus to connect the environmental cues with morphological reactions or plant behaviors, such as hypocotyl elongation. The Pfr form can also be thermodynamically reverted back to Pr through a temperature-dependent process called dark- or thermo-reversion. Increases in temperature between 10 °C and 30 °C accelerate the dark-reversion rate of PHYB, leading to a reduction of the steady-state level of the active Pfr form[19]. This temperature-dependent biophysical property makes PHYB a thermosensor for ambient growth temperatures[19,20].

Extensive studies have demonstrated that PHYB plays a critical role in regulating temperature-dependent hypocotyl growth under SD conditions. Increases in ambient temperature at night promote PHYB's Pfr-to-Pr dark reversion that releases the inhibitory function of PHYB in hypocotyl elongation[19,20]. PHYB controls downstream seedling morphogenesis by regulating a group of basic helix–loop–helix (bHLH) transcriptional regulators, the phytochrome-interacting factors (PIFs)[28]. The PIFs belong to subfamily 15 of the bHLH protein superfamily in *Arabidopsis* and include eight members: PIF1, PIF3–8, and PIL1 (PIF3-like1)[28,29]. Different PIFs play overlapping and distinct roles in *Arabidopsis* seedling morphogenesis[30,31]. For example, hypocotyl elongation is promoted by PIF1, PIF3, PIF4, PIF5, and PIF7[30–33]. However, thermomorphogenesis is centrally controlled by PIF4[9,10]. High temperatures promote the expression of *PIF4* at the transcriptional and posttranslational levels. *PIF4* transcription is tightly regulated by the circadian clock and exhibits distinct rhythmic patterns under LD and SD conditions[16,34]. The transcript level of *PIF4* peaks during the day in LD, but at the end of night in the SD[16,18,34], which coincides with the expression of the growth-relevant PIF4 target genes, particularly genes involved in auxin synthesis and signaling[16,18,35,36]. During early night in SD conditions, *PIF4* transcription is repressed by the evening complex components EARLY FLOWERING3 (ELF3), EARLY FLOWERING4 (ELF4), and LUX ARRHYTHMO (LUX)[37]. This transcriptional inhibition of *PIF4* is released later at night allowing *PIF4* transcripts to peak right before dawn. The transcriptional repression activity of the evening complex is dependent on PHYB and can be negatively regulated by warm temperatures[16,37–39]. Under warm temperatures, the transcript level of *PIF4* is elevated at night, leading to enhanced hypocotyl elongation. At the posttranslational level, PIF4 activity and stability are regulated at night. PIF4 activity is inhibited by ELF3 and TOC1 through direct binding to PIF4's bHLH DNA-binding domain[40,41]. The interaction between PIF4 and PHYB could result in phosphorylation and subsequent ubiquitin/proteasome-dependent degradation of PIF4[42]. Warm temperatures promote PIF4 stabilization[43], which depends on two antagonists for PHYB signaling, DE-ETIOLATED1 (DET1) and CONSTITUTIVE PHOTOMORPHOGENIC1 (COP1)[18,44,45].

Because PHYB's dark-reversion rate can be influenced by temperature in the light, PHYB should theoretically be able to sense temperature during daytime in LD conditions, including continuous light[20]. A growing body of evidence supports a thermosensory role of PHYB under LD conditions. For example, warm-temperature-dependent hypocotyl elongation in LD conditions depends on many PHYB signaling components, including PIF4, COP1, DET1, and HY5[18,46]. However, the function of PHYB signaling in thermosensing in the light remains elusive. One major reason is that the reported temperature responses of hypocotyl elongation in LD and continuous light conditions have been inconsistent[9,14,45]. The minimal responses to warm temperatures under certain LD conditions have left the impression that PHYB plays only a minor role in temperature sensing during daytime[45,47]. In addition, most of the temperature experiments in LD conditions were performed in the white light, where the blue light photoreceptor crytochrome 1 (CRY1) strongly represses warm-temperature-dependent hypocotyl elongation[48]. Hence, PHYB-mediated temperature responses could also be masked by CRY1 signaling in the white light. Here, we show that the warm-temperature response in the white light is unexpectedly complex and largely influenced by the interplay between growth conditions and CRY1 signaling. To circumvent this issue, we characterized the roles of PHYB, PIFs, and HEMERA (HMR)—a PHY-specific signaling component[49]—under monochromatic R light, where only PHYB but not CRY1 is active. Our results demonstrate that PHYB also controls temperature sensing in the light. The daytime temperature sensing by PHYB signals through primarily PIF4 and requires HMR. This study reveals a novel PHYB-mediated temperature-signaling mechanism, in which HMR acts as an essential transcriptional activator to induce the expression of growth-relevant PIF4 target genes and PIF4 accumulation in warm temperatures.

## Results

**PHYB controls thermomorphogenesis in the daytime.** The first step toward understanding PHYB's role in thermosensing under LD or continuous light conditions is to determine an optimal plant growth condition that elicits a significant response to warm temperatures. The reported temperature responses of hypocotyl elongation in LD or continuous light conditions have been inconsistent—the warm-temperature-induced increase in hypocotyl length ranged from 4.7-[40] to merely 0.5-fold[45]. These studies were performed in different experimental settings, in which key variables such as light intensity and seedling age could

contribute to the discrepancies seen in the temperature response. In addition, most of the studies were performed in the white light, where both phytochromes and the blue light photoreceptor CRY1 are activated. The presence of CRY1, which strongly inhibits the warm-temperature response[48], could mask PHYB's thermo-sensory role. To search for a satisfying condition, we devised experiments to determine the variables that could influence the warm-temperature response in the white light. We first examined the hypocotyl responses of Col-0 and *cry1* grown for 4 days at 21 °C and 27 °C under different intensities of white light: 100, 40, and 10 µmol m$^{-2}$ s$^{-1}$. The results of these experiments showed that the warm-temperature response could be altered dramatically under different light intensities: the response declined with increases of light intensity (Fig. 1). The percentage of increase in hypocotyl length at 27 °C vs. 21 °C was 147% in 10 µmol m$^{-2}$ s$^{-1}$, 75% in 40 µmol m$^{-2}$ s$^{-1}$, and only 21% in 100 µmol m$^{-2}$ s$^{-1}$ white light (Fig. 1). More interestingly, the reduced temperature responses in 40 and 100 µmol m$^{-2}$ s$^{-1}$ white light were mainly due to CRY1, because the *cry1* mutant showed 322% and 131% increases in the hypocotyl response under 40 and 100 µmol m$^{-2}$ s$^{-1}$ white light, respectively. These results indicate that CRY1 plays a major role in repressing the warm-temperature response in the white light. However, the role of CRY1 was unexpectedly reversed in 10 µmol m$^{-2}$ s$^{-1}$ white light, where *cry1* was less responsive to the warm temperature compared with Col-0 (Fig. 1), indicating that CRY1 facilitates the warm-temperature response under this low-light condition. The latter result suggests that CRY1's role in temperature sensing depends on growth conditions and it could play completely opposite roles in different light intensities. To test whether the role of CRY1 is also always determined by light intensity, we examined another commonly used growth condition in relatively high light intensity but using relatively older seedlings[9,18,40,46]. Col-0 and *cry1* were grown in

100 µmol m$^{-2}$ s$^{-1}$ white light at 21 °C for 4 days and then were either transferred to 27 °C or maintained at 21 °C under the same light intensity for another 4 days before measurements. In this experimental setup, Col-0 showed a 273% increase in hypocotyl length comparing 27 °C vs. 21 °C (Fig. 1). Surprisingly, *cry1* showed a reduced response of 113% (Fig. 1), suggesting that CRY1 promotes the temperature response under this high-light condition. Therefore, the role of CRY1 is not always determined by light intensity and could also be influenced by seedling's developmental stage. Together, these results show that the temperature response in the white light can be largely influenced by light intensity and developmental stage, as well as the interplay between the growth conditions and CRY1 signaling.

To circumvent the complex effects of CRY1 on the warm-temperature response in the white light, we decided to examine PHYB's role in temperature sensing using monochromatic R light, where only phytochromes, but not cryptochromes, are activated. To that end, we examined the hypocotyl responses of Col-0 at 21 °C and 27 °C under SD, LD, and continuous light conditions with the light periods in monochromatic R light. Under these conditions, the hypocotyl response to warm temperature was more pronounced in continuous R light (Rc) and LD conditions than that in SD in R light. Comparing 27 °C to 21 °C, hypocotyl length increased by 140 ± 18% and 140 ± 6% in Rc and LD conditions, and only 78% ± 5% in SD (Fig. 2a–d). In striking contrast, in Rc and LD conditions, the hypocotyls of *phyB-9* and *phyA-211/phyB-9* mutants—a double mutant of the two prominent phytochromes in *Arabidopsis*—increased only by less than 10% and there was no significant difference between *phyB-9* and *phyA-211/phyB-9*, indicating that the daytime warm-temperature-dependent response in Rc and LD is controlled specifically by PHYB (Fig. 2). It is important to note that *phyB-9* was recently reported to contain a second-site mutation that affects chloroplast development, and this mutation is not present in *phyA-211/phyB-9*[50]. The fact that *phyB-9* and *phyA-211/phyB-9* showed similar temperature responses indicates that the reduction of the temperature response in *phyB-9* is independent of the second-site mutation. Based on these results, we conclude that PHYB also mediates daytime thermosensing in the light.

**Thermomorphogenesis requires the transactivator HMR.** To further investigate how PHYB signaling transduces daytime temperature signals, we tested whether the warm-temperature-induced hypocotyl elongation response in R light requires the phytochrome-signaling component HMR. HMR participates specifically in phytochrome signaling but not CRY signaling in blue light[49]. We have previously shown that HMR is a transcriptional activator involved in early steps of PHYB signaling, including PHYB localization to the subnuclear photobodies and the regulation of the stability and activity of PIF1 and PIF3[49,51,52]. To test whether HMR is involved in thermomorphogenesis, we examined the hypocotyl response of the null *hmr-5* mutant in continuous light, LD, and SD conditions with monochromatic R light under 21 °C and 27 °C. To better compare the temperature response of the mutant, we normalized the relative hypocotyl length differences at 27 °C vs. 21 °C in the mutant against those in Col-0 to calculate the "relative response" for the mutant. The *hmr-5* mutant retained a 39% relative response to warm temperature in SD, but only 25% and 8% relative response in LD and Rc, respectively (Fig. 2), indicating that HMR's role in thermosensing is day-length dependent—it plays a more prominent role in temperature sensing during the day under LD and Rc conditions than at night under SD conditions. Collaborating with the data in R light, *hmr-5* and *hmr-22* in continuous white light almost lost the temperature response with only 2 and 4% relative responses, respectively, these phenotypes were similar to that of

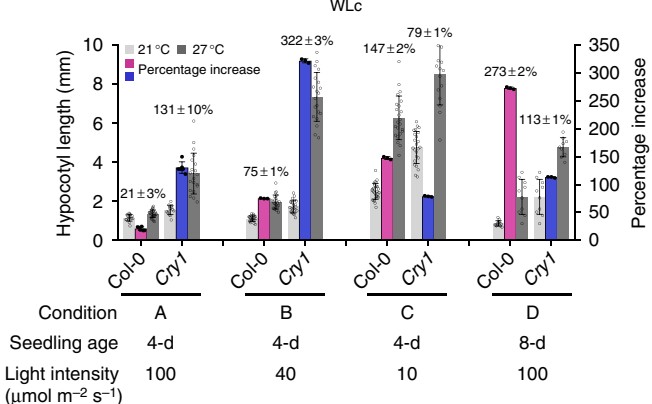

**Fig. 1** Temperature response in white light varies in different growth conditions. Hypocotyl length measurements of Col-0 and *cry1* seedlings in four growth schemes: A, 4-d-old seedlings grown in 100 µmol m$^{-2}$ s$^{-1}$ white light in either 21 °C or 27 °C; B, 4-d-old seedlings grown in 40 µmol m$^{-2}$ s$^{-1}$ white light in either 21 °C or 27 °C; C, 4-d-old seedlings grown in 10 µmol m$^{-2}$ s$^{-1}$ white light in either 21 °C or 27 °C; D, seedlings were grown in 100 µmol m$^{-2}$ s$^{-1}$ white light at 21 °C for 4 days and then either transferred to 27 °C or remained at 21 °C for 4 days before measurements. The hypocotyl response to warm temperatures was measured as the percentage increase in hypocotyl length between 21 °C and 27 °C. The percentages of hypocotyl increases (mean ± SD) at 27 °C are labeled above the columns and represented by the magenta and blue bars for Col-0 and *cry1*, respectively. Error bars for the hypocotyl length represent SD (*n* > 30), errors for the percentage increase represent SD of three to six biological replicates. The underlying source data of the hypocotyl measurements are provided in the Source Data file

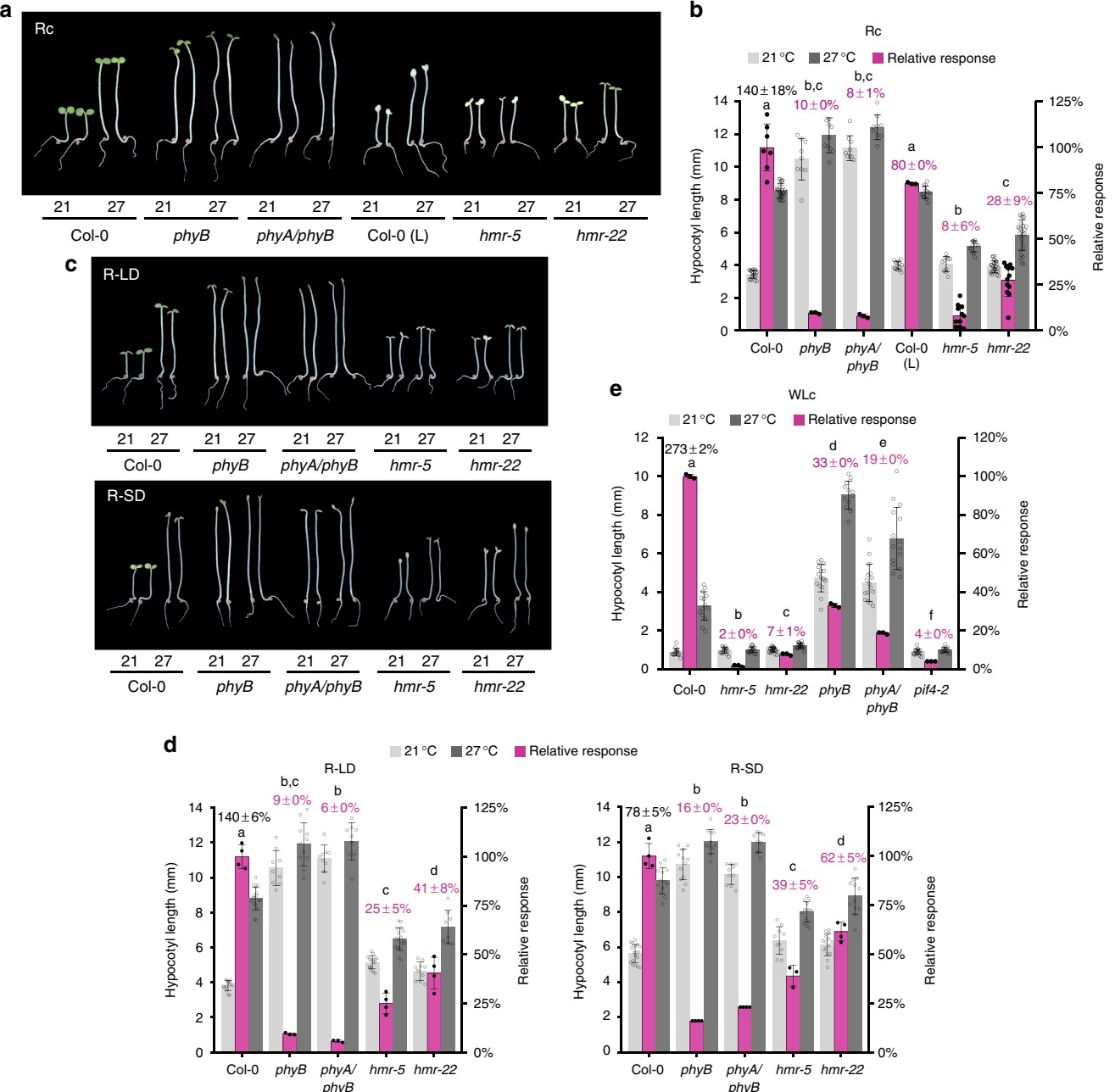

**Fig. 2** Daytime thermosensing is mediated by PHYB and HMR. **a** Images of representative 4-d-old Col-0, *phyB-9*, *phyA-211/phyB-9*, Col-0 treated with 200 µg/ml lincomycin (L), *hmr-5*, and *hmr-22* seedlings grown in 10 µmol m$^{-2}$ s$^{-1}$ R light under 21 °C or 27 °C. **b** Hypocotyl length measurements of seedlings in **a** and their relative responses to the higher temperature. The relative response (magenta bars) is defined as relative hypocotyl response to 27 °C of a mutant compared with that of Col-0. **c** Images of representative 4-d-old Col-0, *phyB-9*, *phyA-211/phyB-9*, *hmr-5*, and *hmr-22* seedlings grown in LD and SD with 10 µmol m$^{-2}$ s$^{-1}$ R light, labeled as R-LD and R-SD, respectively, under 21 °C or 27 °C. **d** Hypocotyl length measurements of seedlings in **c** and their relative responses to warm temperature. **e** Warm-temperature-induced hypocotyl responses of Col-0, *hmr-5*, *hmr-22*, *phyB-9*, *phyA-211/phyB-9*, and *pif4-2* grown in white light. Seedlings were grown in 100 µmol m$^{-2}$ s$^{-1}$ continuous white light (WLc) at 21 °C for 4 days and then either transferred to 27 °C or remained at 21 °C under the same light condition for 4 days before measurements. For **b**, **d**, and **e**, error bars for the hypocotyl length represent SD ($n >$ 30); error bars for the relative responses represent SD of three to ten biological replicates. Black numbers show the mean ± SD values of the absolute responses of Col-0 ($n \geq 3$). Purple numbers show the mean ± SD values of relative responses and different letters denote statistically significant differences in relative responses (ANOVA, Tukey's HSD, $P <$ 0.01, $n \geq 3$). The underlying source data for **b**, **d**, and **e** are provided in the Source Data file

*pif4-2* (Fig. 2e). Under this white light condition, *phyB-9* and *phyA-211/phyB-9* showed much reduced relative responses of 33 and 19% (Fig. 2e), supporting an important role of PHYB for thermosensing in white light. Interestingly, *phyB-9* in the white light showed a greater warm-temperature response than in Rc (Fig. 2b), suggesting that the warm-temperature response in the

white light is also mediated by sensors besides PHYB. Together, these results reveal that HMR is an essential signaling component for PHYB-mediated temperature sensory responses, particularly in the light.

HMR is dual targeted to the nucleus and plastids[49,53]. While nuclear HMR participates in PHY signaling[49,51,52], plastidial

HMR, also called pTAC12 (plastid transcriptionally active 12), is associated with the plastid-encoded plastid RNA polymerase (PEP) and required for chloroplast biogenesis[54,55]. Deficiency of the PEP initiates a plastid-to-nucleus retrograde signaling pathway to modulate nuclear gene expression[56,57], such as the repression of the nuclear-encoded photosynthetic genes as well as the attenuation of the light-dependent inhibition of hypocotyl elongation[58,59]. Therefore, the lack of the warm-temperature-dependent hypocotyl response in *hmr-5* could be due to its chloroplast defects. To test this possibility, we treated Col-0 seedlings with lincomycin—a potent inhibitor for chloroplast translation leading to the loss of the plastid-encoded core subunits of the PEP RNA polymerase[56,60]—and asked whether defects in the PEP and chloroplast biogenesis could alter the warm-temperature response in hypocotyl elongation. The lincomycin-treated albino Col-0 seedlings showed only a minimal effect on the temperature response with an 80% relative response (Fig. 2b). These results indicate that the insensitivity of *hmr-5* to the ambient temperature fluctuation is not caused by defects in the PEP or chloroplast biogenesis, but most likely due to the lack of HMR's function in the nucleus.

HMR participates in early PHY signaling in the nucleus as a transcriptional activator, which regulates the stability and activity of PIF1 and PIF3[49,52]. A weak allele, *hmr-22*, which carries a loss-of-function D516N mutation in HMR's acidic transcriptional activation domain (TAD), abrogates light-dependent degradation of PIF1 and PIF3 as well as the expression of PIF-regulated, growth-relevant genes[52]. The *hmr-22* mutant exhibited reduced hypocotyl responses to warm temperatures in SD, LD, and Rc conditions with 62%, 41%, and 28% relative responses,

respectively (Fig. 2). Consistent with the phenotype of *hmr-5*, *hmr-22*'s temperature responses in Rc and LD were more pronounced than those in SD (Fig. 2). Together, these results provide genetic evidence demonstrating that thermomorphogenesis requires HMR and particularly HMR's TAD.

**HMR mediates thermomorphogenesis through PIF4.** It was surprising that *hmr-5* and *hmr-22* at 27 °C were shorter compared with Col-0, whereas at 21 °C, they were taller than Col-0 (Fig. 2) [49,51,61]. The long-hypocotyl phenotype of *hmr* at 21 °C has been explained by the accumulation of the hypocotyl growth-promoting factors, PIF1 and PIF3[49,51,61]. In contrast, thermomorphogenesis in warm temperature is mediated by the central regulator PIF4[9,10]. Therefore, the discrepancy of *hmr*'s hypocotyl phenotypes under different temperatures might be explained by the distinct PIFs that HMR exerts its functions onto. For example, *hmr*'s short-hypocotyl phenotype at 27 °C might reflect the specific function of HMR to PIF4. To test this hypothesis, we performed genetic experiments to determine the relationship between HMR and individual PIF1, PIF3, PIF4, and PIF5 for hypocotyl elongation at 21 °C and 27 °C. Because the previous studies of PIF functions in thermomorphogenesis were mostly performed under white light[9,18,19,62], these experiments also aimed to confirm the roles of the distinct PIFs in thermomorphogenesis in monochromatic R light. We first re-examined the hypocotyl response of single *pif1*, *pif3*, *pif4*, and *pif5* mutants under 21 °C and 27 °C (Fig. 3a, b). The *pif4-2* mutant showed the least temperature response (Fig. 3a, b). However, under Rc, the *pif4* mutant retained a 39% relative response (Fig. 3b), suggesting

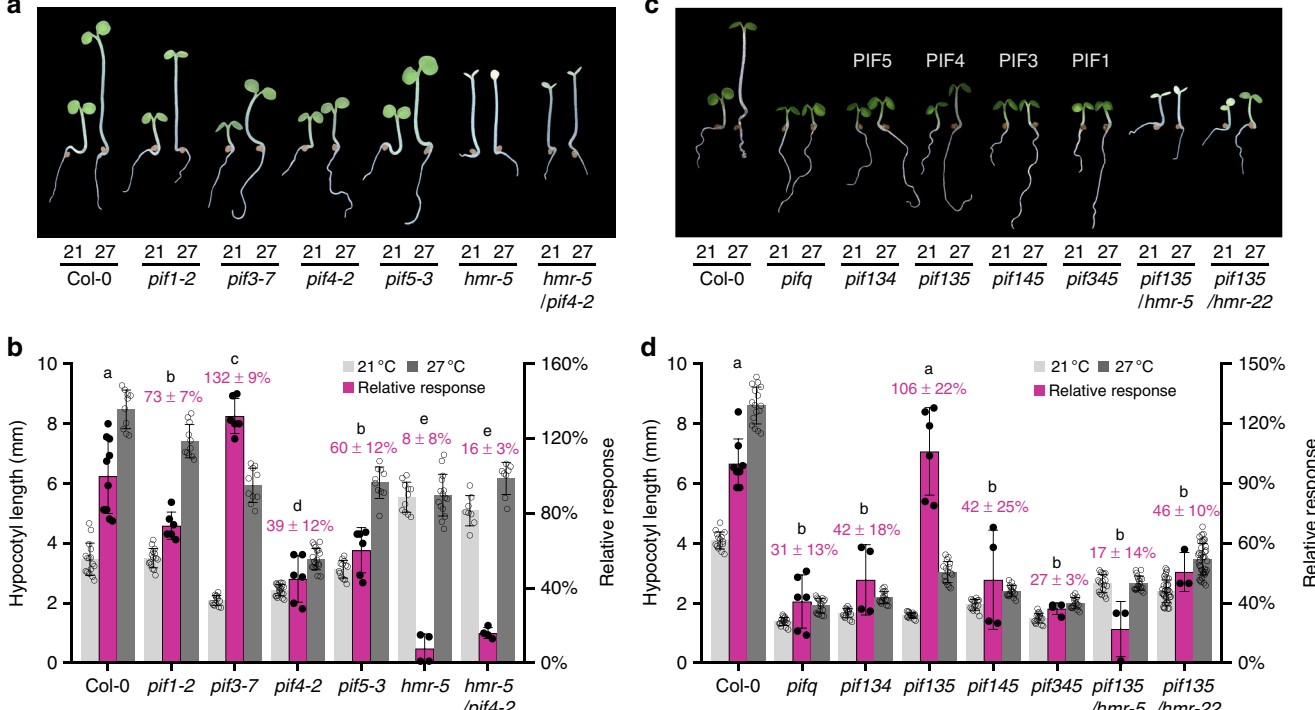

**Fig. 3** Daytime thermosensing signals primarily through a HMR-dependent PIF4 pathway. **a** Images of representative 4-d-old Col-0, *pif1-2*, *pif3-7*, *pif4-2*, *pif5-3*, *hmr-5*, and *hmr-5/pif4-2* seedlings grown in 10 µmol m$^{-2}$ s$^{-1}$ R light under 21 °C or 27 °C. **b** Hypocotyl length measurements of seedlings in **a** and their relative responses to warm temperature. **c** Images of representative 4-d-old Col-0, *pif1345* quadruple mutant (*pifq*), *pif* triple mutants (*pif134*, *pif135*, *pif145*, and *pif345*), *pif135/hmr-5*, and *pif135/hmr-22* seedlings grown in 10 µmol m$^{-2}$ s$^{-1}$ R light under 21 °C or 27 °C. **d** Hypocotyl length measurements of seedlings in **c** and their relative responses to warm temperature. For **b** and **d**, error bars for the hypocotyl length represent SD ($n > 30$); error bars for the relative responses represent SD of three to eight biological replicates. Purple numbers show the mean ± SD values of relative responses and different letters denote statistically significant differences in relative responses (ANOVA, Tukey's HSD, $P < 0.01$, $n \geq 3$). The source data of the hypocotyl measurements in **b** and **d** are provided in the Source Data file

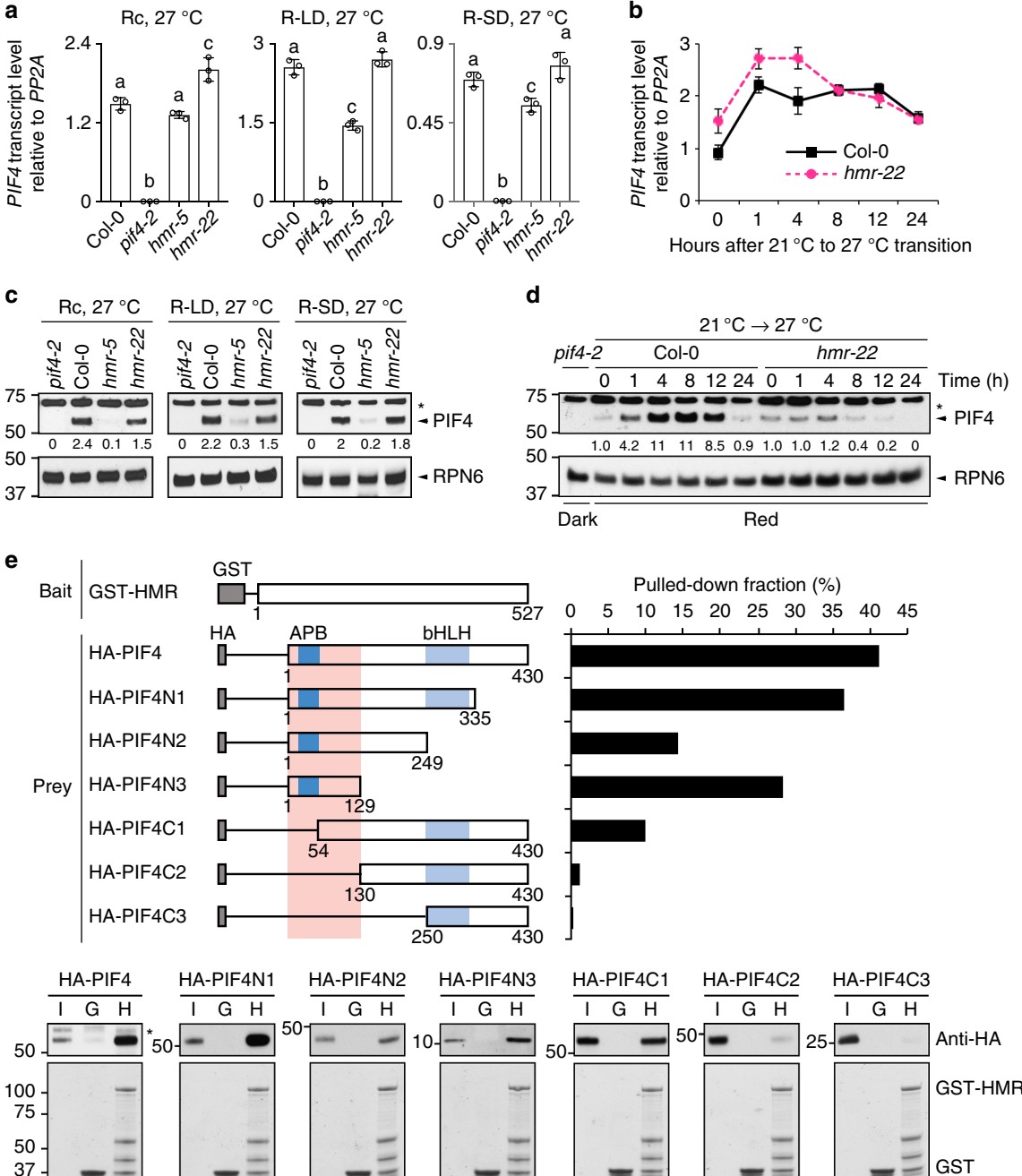

that thermomorphogenesis under monochromatic R light requires other factors besides PIF4. *pif1-2* and *pif5-3* were slightly hyposensitive to warm temperature with 73% and 60% relative responses, respectively, indicating that both PIF1 and PIF5 contribute to the hypocotyl thermoresponse (Fig. 3a, b). Surprisingly, *pif3-7*, despite being shorter than Col-0 at both 21 °C and 27 °C, showed an enhanced response to high temperature with a 132% relative response (Fig. 3b), suggesting that although PIF3 promotes hypocotyl growth, it plays a negative role in the warm-temperature response. Together, these results indicate that day-time thermosensing is mediated primarily through PIF4 with PIF1 and PIF5 playing minor roles, and that the role of PIFs in promoting hypocotyl elongation per se could be separable from those mediating the temperature response.

We then investigated whether individual PIFs were sufficient to mediate thermomorphogenesis by comparing *pifq* with various combinations of *pif* triple mutants that retain only one of the four PIFs. Among the four *pif* triple mutants, only *pif135*, which expresses *PIF4*, exhibited a wild-type thermoresponse, whereas the rest of the *pif* triple mutants behaved similarly to *pifq* (Fig. 3c, d). These results indicate that among the four PIFs, only PIF4 is sufficient to mediate thermomorphogenesis in Rc. Interestingly, the PIF4-dependent thermoresponse was reduced dramatically to 17% in *pif135/hmr-5* and 46% in *pif135/hmr-22* (Fig. 3d). These results demonstrate that the PIF4-dependent warm-temperature response requires HMR.

HMR is required for the PIF4-mediated thermosensory pathway. However, it is worth noting that *hmr-5*, with a relative response of 8%, was significantly more hyposensitive to warm temperature than *pif4-2* with a relative response of 39% (Fig. 3a, b). In addition, the *hmr-5/pif4-2* double mutant showed a relative response similar to *hmr-5* (Fig. 3a, b). These results suggest that

**Fig. 4** HMR interacts with PIF4 and facilitates thermoresponsive PIF4 accumulation. **a** qRT-PCR analyses of the steady-state transcript levels of *PIF4* in 4-d-old Col-0, *pif4-2*, *hmr-5*, and *hmr-22* seedlings grown at 27 °C in Rc, LD (R-LD), and SD (R-SD) conditions with 10 μmol m$^{-2}$ s$^{-1}$ R light. Samples were taken at 96 h after stratification for Rc, 104 h (or ZT 8) for LD, and 94 h (or ZT 22) for SD. Error bars represent SD of three biological replicates. Different letters denote statistically significant differences in *PIF4* transcript levels (ANOVA, Tukey's HSD, $P < 0.01$). **b** qRT-PCR analyses of the transcript levels of *PIF4* in response to elevated temperature. Four-day-old Col-0 and *hmr-22* seedlings grown at 10 μmol m$^{-2}$ s$^{-1}$ Rc at 21 °C were transferred to 27 °C under the same light condition for up to 24 h, and samples were collected at the indicated time points. Error bars represent SD of three replicates. **c** Immunoblot analyses of the PIF4 protein levels in Col-0, *pif4-2*, *hmr-5*, and *hmr-22* under 27 °C in Rc, LD (R-LD), and SD (R-SD) conditions with 10 μmol m$^{-2}$ s$^{-1}$ R light. Samples were taken at 96 h after stratification for Rc, 104 h (or ZT 8) for LD, and 94 h (or ZT 22) for SD. **d** Immunoblot analyses of the PIF4 levels in response to elevated temperature. Four-day-old Col-0 and *hmr-22* seedlings grown at 10 μmol m$^{-2}$ s$^{-1}$ Rc at 21 °C were transferred to 27 °C under the same light condition for up to 24 h and samples were collected at the indicated time points. The dark-grown *pif4-2* sample was used as a negative control. For **c** and **d**, RPN6 was used as a loading control. The relative levels of PIF4, normalized to RPN6, are shown underneath the PIF4 immunoblots. The asterisk indicates nonspecific bands. **e** The N-terminal 1–129 amino acids region of PIF4 is both sufficient and required for the interaction with HMR. GST pull-down assays were performed using *E. coli*-expressed GST-HMR or GST to pull down in vitro-translated HA-tagged full-length PIF4 or PIF4 truncation fragments. The upper-left panel shows the bait and prey constructs used in the GST pull-down assays. The lower panels are immunoblots showing the input and pull-down fractions of HA-tagged PIF4 detected by anti-HA antibodies. The corresponding SDS-PAGE gels show the amounts of immobilized GST or GST-HMR in each assay. The upper-right panel shows the quantification of the amount of immunoprecipitated prey proteins, relative to the total input amount. I, 10% input; G, GST; H, GST-HMR; APB, active phytochrome B-binding motif; bHLH, basic helix–loop–helix domain. The asterisk indicates nonspecific bands. The source data of the qRT-PCR data in **a**, **b** and the immunobolts in **c**–**e** are provided in the Source Data file

HMR participates in PIF4-dependent and PIF4-independent thermosensing pathways. Interestingly, when looking at the *hmr-5* phenotype in *pif4-2* background, *hmr-5/pif4-2* was taller than *pif4-2* at both 21 °C and 27 °C (Fig. 3b). These results indicate that the short-hypocotyl phenotype of *hmr-5* at 27 °C is indeed dependent on PIF4. Therefore, we conclude that daytime thermosensing signals primarily through a HMR-dependent PIF4 pathway.

**HMR facilitates PIF4 accumulation in warm temperatures**. Warm temperature elevates *PIF4* expression at both the transcriptional and posttranscriptional levels[9,43]. We first tested whether HMR was required for the accumulation of *PIF4* transcripts under 27 °C in Rc, LD, and SD conditions as well as during the transition from 21 °C to 27 °C. The *PIF4* transcript level was examined at the 96-h time point after stratification in Rc. Because warm-temperature-mediated induction of *PIF4* is gated by the circadian clock[40,63], we also determined the *PIF4* levels at ZT8 (zeitgeber time) during the day in LD and ZT22 at the end of night in SD conditions, where the *PIF4* transcript level and hypocotyl growth peak in the respective conditions[16,18]. The results of these experiments showed that the steady-state *PIF4* transcript levels in *hmr-5* and *hmr-22* were similar to or within a twofold range compared with that in Col-0 (Fig. 4a). The *PIF4* transcript level was also induced in *hmr-22* during the transition from 21 °C to 27 °C similarly to that in Col-0 (Fig. 4b). These results indicate that HMR does not regulate *PIF4* at the transcript level. In contrast, the steady-state protein level of PIF4 dramatically decreased in *hmr-5* under Rc, LD, and SD conditions compared with that in Col-0 in the respective conditions (Fig. 4c). The level of PIF4 in *hmr-22*, compared with that of Col-0, remained the same in SD conditions but was reduced by 37.5 and 31.8% in Rc and LD conditions, respectively. These results are consistent with the more pronounced phenotypes of *hmr-22* in Rc and LD conditions. Moreover, during the 21 °C to 27 °C transition, *hmr-22* failed to accumulate PIF4 (Fig. 4d), suggesting that the TAD of HMR plays a crucial role in PIF4 accumulation, particularly during the early response to elevated temperatures. Together, these results support the conclusion that HMR and HMR's transactivation activity are required not for warm-temperature-dependent increase in *PIF4* transcript but rather specifically for the thermoresponsive PIF4 protein accumulation.

**HMR interacts directly with PIF4**. We have previously shown that HMR interacts directly with PIF1 and PIF3 to promote their degradation and to activate a group of growth-promoting PIF target genes[52]. HMR binds PIF1 and PIF3 through the active phytochrome B-binding (APB) motif, which is conserved in all PIFs[64], including PIF4[52]. Therefore, we hypothesized that HMR could regulate PIF4 through direct interaction. To test this hypothesis, we performed glutathione S-transferase (GST) pull-down assays using recombinant GST-HMR to pull down in vitro-translated HA-tagged full-length as well as a series of N- or C-terminal truncations of PIF4. These experiments showed that GST-HMR, but not GST alone, can pull down PIF4 (Fig. 4e), confirming that HMR physically interacts with PIF4 in vitro. Moreover, deleting the N-terminal region of PIF4 including amino acids 1–129 in HA-PIF4C2 abolished the interaction with HMR, and this N-terminal fragment alone in HA-PIF4N3 was sufficient to bind HMR (Fig. 4e). Deleting the APB alone in HA-PIF4C1 dramatically dampened the interaction (Fig. 4e). These results indicate that the HMR–PIF4 interaction is mainly mediated by the N-terminus of PIF4 containing the APB motif. We have also tried to detect the in vivo interaction between HMR and PIF4. However, these attempts were unsuccessful. This might be due to low levels of HMR in the nucleus in both Col-0 and the transgenic lines expressing HMR-HA[51,52] or the HMR–PIF4 interaction being transient therefore making it difficult to capture.

**Thermoresponsive genes are activated by HMR's TAD**. We then examined how the temperature-responsive PIF4 target genes are affected in the *hmr* mutants. To that end, we focused on three well-characterized warm-temperature-induced marker genes involved in the biosynthesis and signaling of the plant growth hormone auxin—*YUC8*, *IAA19*, and *IAA29*[36]. In Rc and LD conditions at 27 °C, the steady-state levels of *YUC8*, *IAA19*, and *IAA29* in *hmr-5* were dramatically reduced to levels similar to those in *pif4-2* (Fig. 5a). However, the expression of these genes was only moderately reduced in SD (Fig. 5a). These results are consistent with the notion that HMR plays a more important role for thermosensing in LD and Rc conditions. The expression of these genes was only slightly changed in *hmr-22* (Fig. 5a). However, during the 21 °C to 27 °C transition, the induction of all three marker genes was blocked in *hmr-22* (Fig. 5b). The latter results indicate that the activity of HMR's TAD is required for the activation of PIF4 target genes during the transition to warm temperatures.

**Fusing VP16 TAD to HMR22 rescues its signaling functions**. It was intriguing that a loss-of-function mutation of HMR's TAD in

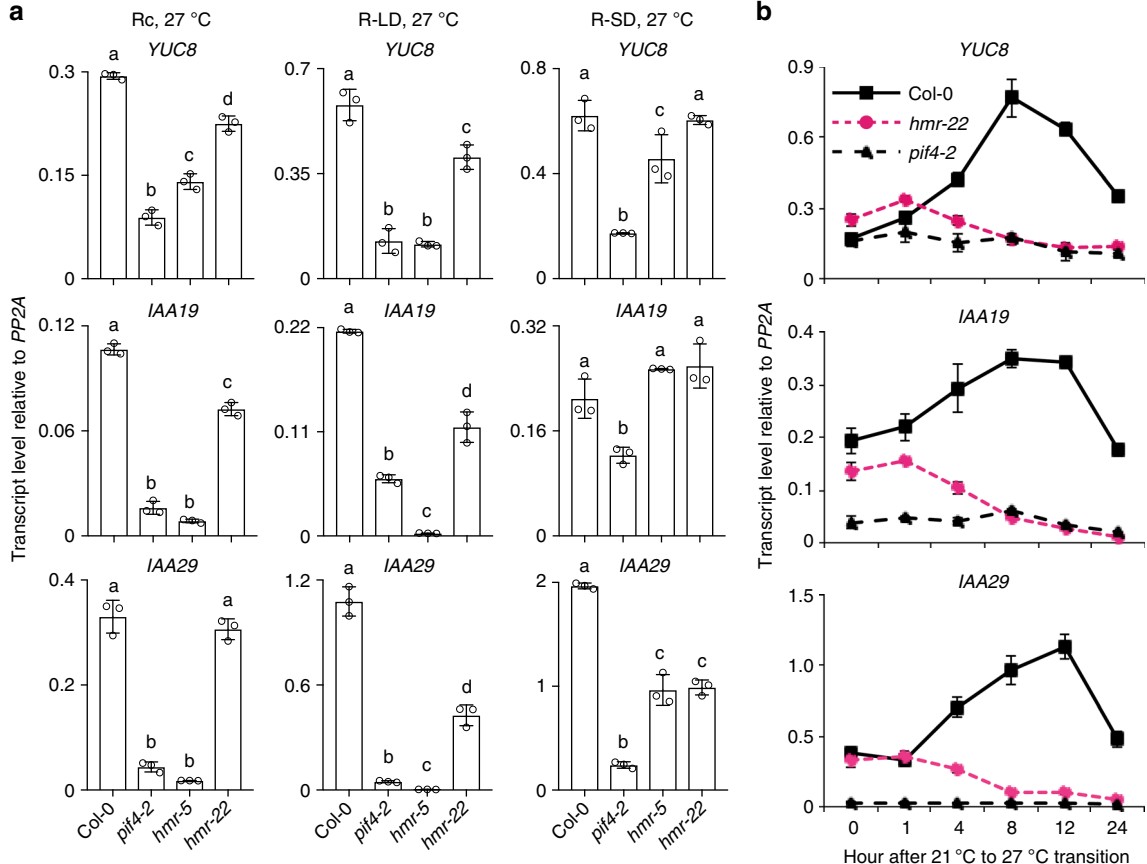

**Fig. 5** HMR's TAD is required for the activation of thermoresponsive genes. **a** qRT-PCR analyses of the steady-state levels of warm-temperature-induced PIF4 direct target genes, *YUC8*, *IAA29*, and *IAA19*, in 4-d-old Col-0, *pif4-2*, *hmr-5*, and *hmr-22* seedlings grown under 27 °C in Rc, LD (R-LD), and SD (R-SD) conditions with 10 µmol m$^{-2}$ s$^{-1}$ R light. Samples were taken at 96 h after stratification for Rc, 104 h (or ZT 8) for LD, and 94 h (or ZT 22) for SD. Error bars represent SD of three replicates. Different letters denote statistically significant differences in transcript levels (ANOVA, Tukey's HSD, *P* < 0.01). **b** qRT-PCR analyses of the transcript levels of *YUC8*, *IAA29*, and *IAA19* in response to elevated temperatures in Col-0, *pif4-2*, and *hmr-22*. Four-day-old Col-0, *pif4-2*, and *hmr-22* seedlings grown at 10 µmol m$^{-2}$ s$^{-1}$ Rc at 21 °C were transferred to 27 °C under the same light condition for up to 24 h, and samples were collected at the indicated time points. Error bars represent SD of three replicates. For all the qRT-PCR analyses, the transcript levels were calculated relative to those of *PP2A*. The source data of the qRT-PCR data in **a** and **b** are provided in the Source Data file

*hmr-22* could lead to defects in both the activation of PIF4 target genes as well as PIF4 accumulation. This raises the possibility that PIF4 accumulation is dependent on the activation of its target genes by HMR. To test this hypothesis and to confirm that the in vivo function of HMR relies on the activity of its TAD, we fused the transactivation domain of VP16 and an HA tag to HMR$^{D516N}$ (hereafter named HMR22), and asked whether HMR22-HA-VP16 could rescue the defects of HMR22. To this end, we first examined whether *HMR22-HA-VP16/hmr-22* lines could rescue the hypocotyl growth defect of *hmr-22* in response to warm temperature. Indeed, while expressing HMR-HA in *hmr-22* rescued the relative response of *hmr-22* from 27 to 100%, HMR22-HA-VP16 rescued to 75% relative response (Fig. 6a, b). Consistent with the hypocotyl phenotype, the expression of the three thermoresponsive PIF4 target genes in *HMR22-HA-VP16/hmr-22* was also elevated to their levels in *HMR-HA/hmr-22* (Fig. 6c). These data indicate that the transactivation domain of VP16 can rescue the defect of HMR's TAD, supporting the notion that the thermoresponsive genes are activated by HMR's TAD in vivo. More interestingly, during the 21 °C to 27 °C transition, the *HMR22-HA-VP16/hmr-22* line also largely rescued *hmr-22*'s defect in PIF4 accumulation (Fig. 6d). These results provide genetic evidence that PIF4 accumulation in warm temperature is dependent on HMR and particularly HMR's transactivation activity in vivo.

## Discussion

Plants are more likely to encounter higher temperatures during the daytime in the light. Therefore, the understanding of daytime temperature sensing at the molecular level will be critical in deciphering how plants sense, respond, and adapt to higher temperatures. However, if and how PHYB signaling participates in temperature sensing in the light remains elusive. We show here that the complex temperature responses in the white light, mainly contributed by the interplay between growth conditions and CRY1 signaling (Fig. 1), have made it difficult to discern the function of PHYB in thermonsensing. Using monochromatic R light conditions specifically activating the phytochrome photoreceptors, we show that PHYB controls the daytime hypocotyl elongation response to warm temperatures in LD and continuous light conditions. Daytime temperature sensing requires the phytochrome-specific signaling component HMR and its transcriptional activation function for regulating the activity and stability of the central temperature-responsive transcription factor PIF4. We propose that while PHYB mediates night-time temperature responses in SD conditions by controlling *PIF4* transcription through the evening complex[37–39], PHYB controls daytime temperature-dependent hypocotyl elongation in the light mainly through the regulation of the activity and stability of PIF4 by HMR.

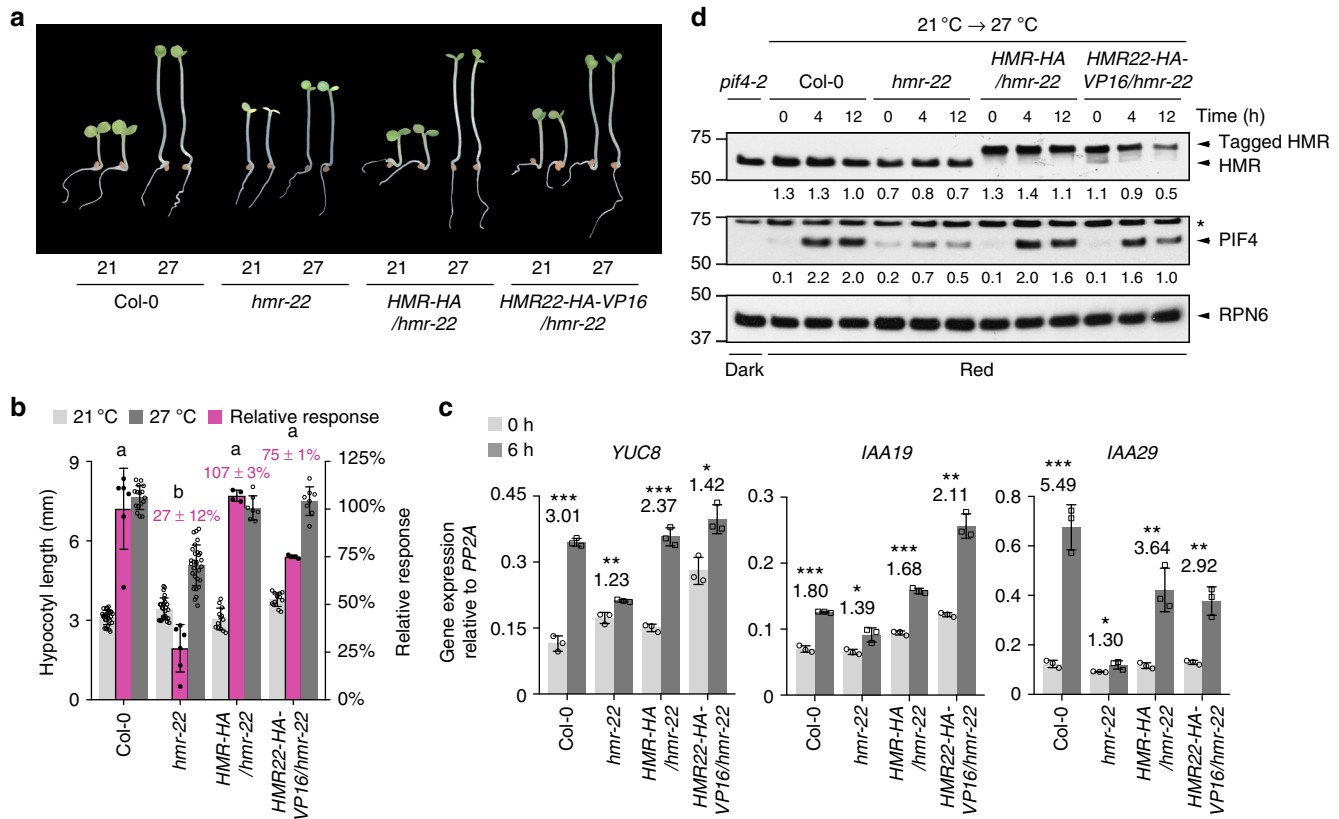

**Fig. 6** Expressing HMR22 fused with a VP16 TAD rescued the thermoresponsive defects of *hmr-22*. **a** Images of representative 4-d-old Col-0, *hmr-22*, *HMR-HA/hmr-22*, and *HMR22-HA-VP16/hmr-22* seedlings grown at 10 μmol m$^{-2}$ s$^{-1}$ R light under 21 °C or 27 °C. **b** Hypocotyl length measurements of seedlings in **a** and their relative responses to warm temperature. Error bars for the hypocotyl length represent SD ($n > 30$); error bars for the relative responses represent SD of two to six biological replicates. Purple numbers show the mean ± SD values of relative responses and different letters denote statistically significant differences in relative responses (ANOVA, Tukey's HSD, $P < 0.01$, $n \geq 2$). **c** qRT-PCR analyses of the transcript levels of *YUC8*, *IAA29*, and *IAA19* in response to elevated temperature in 4-d-old Col-0, *hmr-22*, *HMR-HA/hmr-22*, and *HMR22-HA-VP16/hmr-22*. Seedlings were grown at 10 μmol m$^{-2}$ s$^{-1}$ R light for 96 h after stratification and then transferred to 27 °C for 6 h. Samples were taken at 0- and 6-h time points. Error bars represent SD of three replicates. For all the qRT-PCR analyses, the transcript levels were calculated relative to those of *PP2A*. Numbers indicate fold changes between 21 °C and 27 °C. The statistical significance was analyzed by Student's t-test (*$P < 0.05$, **$P < 0.01$, ***$P < 0.001$). **d** Immunoblot analyses of the PIF4 level in response to elevated temperature. Four-day-old Col-0, *hmr-22*, *HMR-HA/hmr-22*, and *HMR22-HA-VP16/hmr-22* seedlings grown at 10 μmol m$^{-2}$ s$^{-1}$ R light at 21 °C were transferred to 27 °C under the same light condition, and samples were collected at the indicated time points. The dark-grown *pif4-2* sample was used as a negative control. RPN6 was used as a loading control. The relative levels of HMR and PIF4, normalized to RPN6, are shown underneath the corresponding immunoblots. The asterisk indicates nonspecific bands. The source data of the immunoblots in **d**, the hypocotyl measurements in **b**, and the qRT-PCR data in **c** are provided in the Source Data file

Hypocotyl elongation occurs during the daytime or in the light under LD conditions and before dawn or in the dark in SD[15,16,18]. The main difference between the LD and SD hypocotyl elongation is that daytime hypocotyl elongation is regulated directly by active PHYB, whereas night-time elongation in SD is indirectly controlled by PHYB through the regulation of *PIF4* transcription via the evening complex. Therefore, the LD and SD hypocotyl elongation represents two distinct growth modes—PHYB-dependent growth in the light and PHYB-independent growth in the dark, respectively. Our results demonstrate that PHYB plays an equally important role in daytime temperature sensing (Fig. 2). PHYB's thermosensory role during daytime has been poorly recognized, largely because of the inconsistent hypocotyl responses to warm temperatures in LD conditions. The minimal temperature response in hypocotyl elongation under certain LD conditions in the white light[45] has led to the notion that PHYB's temperature-dependent dark-reversion rate operates primarily in the dark by influencing how fast the active PHYB diminishes at night[19,47,65]. However, an acceleration in dark-reversion rate, similar to an increase of FR light, is expected to tip the balance of

the Pfr to Pr ratio as well as increase the processivity or the cycling rate between Pr and Pfr of PHYB in the light[20]. Therefore, PHYB should theoretically be able to perceive ambient temperature changes in the light[20]. In support of this conclusion, we provide genetic evidence that in monochromatic R light, the warm-temperature-dependent hypocotyl response is more pronounced in Rc and LD than SD conditions, and the temperature response in Rc and LD conditions is PHYB-dependent (Fig. 2). These results, combined with the fact that the temperature-dependent hypocotyl elongation occurs during daytime in the light[15–18], indicate that PHYB controls thermomorphogenesis in the light. Our results also show that PHYB plays a major role in continuous white light (Fig. 2e), although it is worth noting that under our white light condition, *phyB-9* retained 33% of Col-0's warm-temperature response, suggesting that temperature sensing in the white light must also be mediated by other sensors besides PHYB. It is also intriguing that the PHYB-dependent hypocotyl thermoresponse is masked by CRY1 under certain growth conditions but promoted under others (Fig. 1). The inhibitory role of CRY1 to the temperature responses is likely through its dominant

inhibitory function on PIF4[48]. Further investigations are needed to dissect the dynamic interactions between PHYB and CRY1 pathways in thermomorphogenesis.

Supporting a critical role of PHYB signaling in thermosensing in the light, we showed that daytime thermosensing requires a phytochrome-specific signaling component HMR. HMR contributes more significantly to thermonsensing in Rc and LD than in SD conditions (Fig. 2). This is consistent with the published results that HMR's function is PHYB-dependent and is required for hypocotyl growth in the light but not in the dark[49,51]. Other signaling components have also been shown to exert significant roles only under LD or SD conditions. For example, HY5 plays a more pronounced role in thermorsensing in LD conditions[45,46], whereas ELF3 is only involved in SD conditions[18]. Together, our results, in combination with the published data, strongly support the notion that PHYB controls thermo-responsive hypocotyl elongation under LD and SD through distinct mechanisms. We propose that daytime PHYB-mediated thermosensing is mainly through the regulation of the activity and stability of PIF4 through HMR.

A central mechanism underlying thermomorphogenesis in both LD and SD conditions is through the regulation of the central temperature-responsive transcriptional regulator PIF4 and the activation of PIF4-dependent auxin biosynthetic and signaling genes, including *YUC8*, *IAA19*, and *IAA29*[9,10,18,36]. This study reveals that the PIF4-mediated thermosensing depends on HMR. This conclusion is solidly supported by the genetic evidence that both *hmr-5* and *hmr-22* are defective in the warm-temperature-dependent hypocotyl response (Fig. 2). Knocking out *PIF1*, *PIF3*, and *PIF5*, with only *PIF4* present, the PIF4-dependent thermo-response in *pif135* was abolished in the *pif135/hmr-5* mutant, demonstrating that the function of PIF4 in thermomorphogenesis depends on HMR (Fig. 3c). Our results show that HMR regulates PIF4 not at the transcript level but rather at the posttranslational level by controlling the expression of the temperature-responsive PIF4 target genes and PIF4 accumulation. HMR is a transcriptional activator interacting directly with PIF1 and PIF3[52]. It has been shown that HMR's acidic TAD is responsible for the activation of PIF-regulated, growth-relevant genes as well as the degradation of PIF1 and PIF3 in the light at 21 °C[52]. Here, we show that HMR interacts with PIF4 in vitro through PIF4's N-terminal region (Fig. 4e). We were not able to detect in vivo HMR–PIF4 interaction by immunoprecipitation, which might be due to low levels of HMR in the nucleus or the dynamic nature of the interaction. However, our results provide genetic evidence that defects in the transactivation function of HMR's TAD in *hmr-22* impair the warm-temperature-dependent induction of PIF4 target genes (Fig. 5b) and PIF4 accumulation (Fig. 4d). More interestingly, fusing the transactivation domain of VP16 to HMR22 rescued both defects of *hmr-22* (Fig. 6c, d), indicating a causal relationship between the activation of the temperature-responsive genes by HMR and PIF4 accumulation. It is still unclear how the activity of HMR's TAD can stabilize PIF4. One possibility is that the TAD of HMR facilitates the formation of the preinitiation complex, which helps to prevent PIF4 from degradation by the ubiquitin–proteasome pathway. PIF4 degradation is mediated by CULLIN3-based E3 ubiquitin ligases with BLADE-ON-PETIOLE (BOP) 1 and 2 as the substrate recognition subunit[66]. Mutants of the *BOP* genes showed an enhanced hypocotyl response to warm temperature[66]. One possibility is that HMR's TAD function blocks PIF4 from the BOP-dependent PIF4 degradation. Warm-temperature-dependent PIF4 accumulation also depends on DET1 and COP1, both of which are also required for the induction of PIF4 target genes[18,44,45]. HMR could work with DET1 and COP1 in promoting PIF4 accumulation in higher temperatures.

In contrast to the white light conditions, where thermomorphogenesis relies almost entirely on PIF4[9,18,62], in R light, the hypocotyl response to warm temperature also depends on at least PIF1 and PIF5 (Fig. 3a, b). Interestingly, PIF3, albeit its role in promoting hypocotyl elongation, negatively regulates the hypocotyl response to warm temperature (Fig. 3a, b). Our genetic data indicate that HMR participates in both PIF4-dependent and PIF4-independent pathways in thermomorphogenesis, because both *hmr-5* and *hmr-5/pif4* were more hyposensitive to warm temperatures than *pif4* (Fig. 3a, b). Given that HMR interacts with all PIFs[52], these data might suggest that HMR could also regulate PIF1 and PIF5 in thermosensing.

It is intriguing that HMR showed opposite effects to the stability of PIF3 and PIF4. The TAD of HMR promotes PIF3 degradation but PIF4 accumulation (Fig. 4d)[52]. These distinct functions of HMR on the stability of PIF3 and PIF4 provide an explanation for the contrasting phenotypes of the *hmr* mutants under 21 °C and 27 °C: the long-hypocotyl phenotype of *hmr* mutants at 21 °C could be mainly due to the accumulation of PIF3[52], whereas the short-hypocotyl phenotype at 27 °C is caused by the defect in PIF4 accumulation (Fig. 3a). Supporting these conclusions, in the *pif4* background, the *hmr-5/pif4* double mutant became taller than *pif4* (Fig. 3a), demonstrating that the short-hypocotyl phenotype of *hmr-5* in warm temperatures is PIF4-dependent. Therefore, although HMR interacts with PIF3 and PIF4, and both of PIF3 and PIF4 are regulated by HMR's TAD, the modes of action of HMR on PIF3 and PIF4 stability seem to be different. This discrepancy may be explained by the distinct degradation mechanisms of PIF3 and PIF4. For example, the degradation of PIF3, but not PIF4, is mediated by the carboxy-terminal signaling output module of PHYB[61]. PIF3 degradation is mediated by the Cullin3-LRB [(Light-Response Bric-a-Brack/Tramtrack/Broad (BTB)] and Cullin1-EBF (EIN3-binding F box protein) E3 ubiquitin ligases[67,68], whereas PIF4 degradation is mediated by the Cullin3-BOP E3 ubiquitin ligases[66]. Further work will be focused on elucidating the mechanistic links between HMR's TAD to the regulation of PIF3 and PIF4 stability.

HMR is dual targeted to plastids and the nucleus[49,53]. The plastidial HMR, also called pTAC12, is an essential component of the PEP and is required for the induction of the plastid-encoded photosynthetic genes[54]. Because phytochrome signaling in the nucleus can be controlled by chloroplast-to-nucleus or retrograde signals, particularly from the activity of the PEP[57], one challenge in defining HMR's function is to dissect the contributions of the nuclear and plastidial HMR. Our results show that hypocotyl elongation in warm temperatures was intact when plastid translation was inhibited by lincomycin (Fig. 2a). Because a major effect of lincomycin is to eliminate the expression of the core subunits of the PEP, which triggers the GUN signaling[57], our results indicate that the hypocotyl response to warm temperatures is not influenced by the loss of the PEP or GUN signaling. Therefore, the temperature phenotype of *hmr* is due to defects of the nuclear HMR. These results are consistent with our previous findings that the hypocotyl phenotype of *hmr* at 21 °C is caused by PIF1 and PIF3 accumulation in the nucleus, which is separable from the chloroplast defects[52], and HMR-regulated, growth-relevant genes are not regulated by retrograde signaling[53]. Together, these results provide genetic evidence that reveals HMR as an essential regulator for the nuclear PHYB- and PIF4-mediated thermomorphogenesis mechanism in the light.

Photobodies are PHYB-containing light-sensory subnuclear domains that are associated with early signaling events, such as the regulation of PIF3 degradation[61,65,69–72]. The biogenesis of photobodies is regulated directly by light quality and quantity, and therefore is determined by the Pfr form of PHYB[73,74]. In SD

conditions, PHYB could regulate the evening complex and *PIF4* transcription through the photobodies[20,69]. Photobodies function to stabilize the Pfr form of PHYB, therefore inhibiting *PIF4* transcription in the dark[65,75,76]. Also, photobodies could directly contribute to PHYB's function in the regulation of the evening complex, because the evening complex components are constituents of the photobodies[77], and the evening complex and PHYB bind to shared genome-wide gene-regulatory sites[38]. During the daytime, the morphology, including size and number, of photobodies is regulated by light and temperature[20,73]. HMR was identified as a light-signaling component required for the formation of large photobodies[49]. Given the critical function of HMR in thermomorphogenesis, this study provides genetic evidence supporting a role of photobodies in thermosensing in the daytime, which is consistent with the dynamic changes of photobody morphology under different temperatures[20].

In conclusion, this study demonstrates a thermosensing role of PHYB in the daytime. We have identified HMR as an essential transcription activator in thermomorphogenesis. Our results support a novel PHYB-mediated temperature-signaling mechanism, in which HMR's TAD facilitates the activation of thermoresponsive PIF4 target genes as well as PIF4 accumulation. Future investigations will focus on understanding the mechanistic link between the activity of HMR's TAD and PIF4 accumulation, as well as the interaction between PHYB and CRY1 signaling in thermomorphogenesis in the white light.

## Methods

**Plant materials and growth conditions**. The Columbia ecotype (Col-0) of *Arabidopsis* was used throughout this study. The *phyB-9*, *phyA-211/phyB-9*, *hmr-5*, and *hmr-22* mutants, as well as transgenic lines *HMR-HA/hmr-22* and *HMR22-HA-VP16/hmr-22* were previously described[52]. Because *hmr-5* is albino and seedling lethal, homozygous *hmr-5* seedlings used in this study were from segregating populations. Single, triple, and quadruple *pif* mutants, including *pif1-2* (SALK_072677), *pif3-3* (CS66042), *pif4-2* (SAIL_1288_E07), *pif5-3* (SALK_087012), *pif134* (CS66500), *pif135* (CS66047), *pif145* (CS68095), *pif345* (CS66048), and *pifq* were previously described[30] and obtained from Arabidopsis Biological Resource Center. The *cry1* mutant was reported as *hy4-B104*[78]. Seeds were surface sterilized[51] and plated on half-strength Murashige and Skoog media with Gamborg's vitamins (MSP0506, Caisson Laboratories, North Logan, UT, USA), 0.5 mM MES (pH 5.7), and 0.8% (w/v) agar (A038, Caisson Laboratories, North Logan, UT, USA). For lincomycin treatments, media were supplemented with 220 μg/ml lincomycin hydrochloride (L2774, Sigma-Aldrich). Seeds were stratified in the dark at 4 °C for 5 days before treatment of specific light and temperature in an LED chamber (Percival Scientific). R light was kept at 10 μmol m$^{-2}$ s$^{-1}$ for Rc, SD (8 h day/16 h night), or LD (16 h day/8 h night) conditions. Fluence rates of light were measured using an Apogee PS200 spectroradiometer (Apogee Instruments Inc., Logan, UT, USA).

**Hypocotyl measurement**. Seedlings were scanned using an Epson Perfection V700 photo scanner, and at least 30 hypocotyls were measured for each genotype and treatment using NIH ImageJ software (http://rsb.info.nih.gov/nih-image/).

**Protein extraction and immunoblots**. For total protein extraction, *Arabidopsis* seedlings were harvested and ground using a Mini-Beadbeater-24 (BioSpec Products, Inc.) in three volumes (mg/μL) of extraction buffer containing 100 mM Tris-Cl pH 7.5, 100 mM NaCl, 5 mM EDTA, 1% SDS, 5 mM DTT, 10 mM β-mercaptoethanol, 40 μM MG115 (Sigma-Aldrich), 40 μM MG132 (Sigma-Aldrich), 1× phosphatase inhibitor cocktail 3 (Sigma-Aldrich), 1× EDTA-free protease inhibitor cocktail (Roche), and 0.01% bromophenol blue. Samples were immediately boiled for 10 min and centrifuged at 16,000×*g* for 10 min. Proteins in the supernatant were separated by SDS-PAGE, transferred to nitrocellulose membranes, probed with the indicated primary antibodies, and then incubated with horseradish peroxidase-conjugated anti-goat, anti-mouse, or anti-rabbit secondary antibodies (Bio-Rad). Primary antibodies, including monoclonal mouse anti-HA antibodies (Sigma-Aldrich, H3663), polyclonal goat anti-HA antibodies (Genscript, A00168), polyclonal rabbit anti-HMR antibodies (homemade), polyclonal rabbit anti-PIF4 antibodies (Agrisera, AS 12 1860), and polyclonal rabbit anti-RPN6 antibodies (Enzo Life Sciences, BML-PW8370-0100) were used at 1:1000 dilution. Signals were detected by chemiluminescence using a SuperSignal kit (ThermoFisher Scientific).

**RNA extraction and quantitative reverse transcription-PCR**. Seedlings for RNA extraction were collected, frozen in liquid nitrogen, and stored at −80 °C before processing. Samples were ground in liquid nitrogen, and RNA was extracted using the Quick-RNA MiniPrep kit with on-column DNase I digestion (Zymo Research). cDNA synthesis was performed with the Superscript II First Strand cDNA Synthesis Kit (ThermoFisher Scientific). For qRT-PCR, cDNA was mixed with iQ SYBR Green Supermix (Bio-Rad) and primers (Supplementary Table 1). qRT-PCR reactions were performed in triplicate with a Bio-Rad CFX Connect Real-Time PCR Detection System.

**Plasmid construction**. To make the bait vector for GST pull-down assays, the full-length coding sequence of HMR was inserted into EcoRI and PstI sites of pET42b (Novagen). To make the prey vectors, sequences encoding PIF4 full length (amino acids 1–430) and fragments (amino acids 1–335, 1–249, 1–129, 54–430, 130–430, and 250–430) were inserted into EcoRI and XmaI sites of pCMX-PL2-NterHA. All the primers used for making constructs are listed in Supplementary Table 2.

**GST pull-down assays**. Bait proteins (GST and GST-HMR) were expressed in the *E. coli* BL21 (DE3) strain. Cells were harvested by centrifugation, and the pellet was resuspended in E buffer containing 50 mM Tris-Cl (pH 7.5), 100 mM NaCl, 1 mM EDTA, 1 mM EGTA, 1% DMSO, 2 mM DTT, and bacterial protease inhibitor cocktail (Sigma-Aldrich). All subsequent purification and binding steps were carried out at 4 °C. Cells were lysed by French press, and the lysate was centrifuged at 13,000×*g* for 20 min. Proteins were then precipitated with 3.3 M ammonium sulfate and incubated for 4 h at 4 °C. After centrifugation at 10,000×*g* for 30 min, protein pellets were resuspended in E buffer. Insoluble protein was removed by centrifugation at 13,000×*g* for 1 h, and the supernatant was dialyzed against E buffer overnight at 4 °C.

To immobilize bait proteins, protein extracts were incubated with glutathione Sepharose beads (GE Healthcare) equilibrated in E buffer for 2 h, and then beads were washed four times in E buffer supplemented with 0.1% Nonidet P-40. HA-tagged prey proteins were produced in vitro using the TNT T7-Coupled Reticulocyte Lysate System (Promega) according to the manufacturer's protocol, and with the above-mentioned plasmids[52]. TNT products were then diluted in E buffer with 0.1% Nonidet P-40 and incubated with bead-immobilized bait proteins at 4 °C for 2 h. After binding, beads were washed four times in E buffer with 0.1% Nonidet P-40, and then protein was eluted by boiling in 1× Laemmli sample buffer. Bound proteins were separated by SDS-PAGE, and prey proteins were detected with immunoblots using polyclonal goat anti-HA antibodies (Genescript). Bait proteins were visualized by staining SDS-PAGE gels with Coomassie Brilliant blue.

**Reporting Summary**. Further information on experimental design is available in the Nature Research Reporting Summary linked to this article.

## Data availability

*Arabidopsis* mutants and transgenic lines, as well as plasmids generated during the current study are available from the corresponding author on reasonable request. The source data of the gels and immunoblots in Figs. 4c–e and 6d as well as the source data underlying Figs. 1, 2b, 2d, 2e, 3b, 3d, 4a, b, 51-b, and 6b, c are provided in the Source Data file.

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

## Acknowledgements
We thank Drs. Peter Quail (UC Berkeley) and Chentao Lin (UC Los Angeles) for kindly providing the *pifq* and *cry1* mutants, respectively. We thank Dr. Chan Yul Yoo, Dr. Keunhwa Kim, and Joseph Hahm for their critical comments and suggestions regarding the manuscript. This work was supported by grant R01GM087388 from the National Institute of General Medical Sciences to M.C.

## Author contributions
Y.Q. and M.C. conceived the original research plan; M.C. supervised the experiments; Y.Q., M.L., R.J.K., and C.M. performed the experiments; Y.Q. and M.C. analyzed the data; Y.Q. and M.C. wrote the article.

## Additional information

**Competing interests:** The authors declare no competing interests.

**Journal Peer Review Information:** *Nature Communications* thanks the anonymous reviewers for their contribution to the peer review of this work. Peer reviewer reports are available.

