## [Peer Review File · Nature Communications]

Reviewers' comments:

Reviewer #1 (Remarks to the Author):

The authors have previously shown that HMR plays a role in phyB downstream signalling. In the present manuscript they extend the role of HMR and show that it is involved in phyB mediated temperature-regulation of hypocotyl growth by controlling PIF4 accumulation. Overall, this is an interesting manuscript and I enjoyed reading it. Below I have a couple of comments and questions:

Abstract: The authors point out that temperature sensing by phyB is thought to operate mainly at night. However, I think it is worth mentioning that a function of phyB in daytime temperature sensing has been suggested as well and that even in light, temperature affects the levels of active phyB (Legris et al., 2016, Science, 354, 897-900: "...Temperature affects the Pfr status of phyB mainly via kr1 in the light (Fig. 1) and via kr2 during the night (19).")

Abstract: "Fusing the VP16 TAD..."  only when reading the full manuscript it became clear to me that the VP16 TAD was fused to the HMR22 mutant protein  please reword so that this is clear from the abstract.

Introduction: "It has been thought that PHYB plays a minor role in temperature-sensing during daytime 45,46"  I agree that this has been assumed but reference 19 shows that temperature affects the Pfr status of phyB in light; thus, there has been evidence that phyB also works as temperature sensor during daytime.

Results: "We hypothesized that the daytime thermosensory role of PHYB in LD conditions might be masked by the blue light photoreceptor CRY1 in the white light."  to test this hypothesis the authors performed experiments in red light, which is fine. However, an additional experiment would be to use the cry1 mutants in white light, which would further strengthen their conclusions.

Results: "... under SD, LD, and continuous light conditions with monochromatic R light."  from this it is not clear whether they also used red light for the LD and SD treatments. Based on the figure legend I think they also used red light for SD and LD  please reword in the result section so that this is clear.

Results: "These results support our hypothesis that the daytime hypocotyl response is masked in the white light."  If I understand correctly, the authors performed all their experiments in cR or in LD/SD with R/D cycles. To make the comparison to white light, I think it is essential to include the white light control in their experiments, i.e. comparing their own red light data to white light data in the literature is problematic since differences between their own experimental setup and the setup in the literature not related to red/white light might affect seedling growth.

Results: "... we examined the hypocotyl response of the null hmr-5 mutant in continuous light Rc"  I thought that hmr null mutants are seedling lethal ... if this is true, how could they measure hypocotyl growth? Using a segregating population, sorting out seedlings with a WT phenotype, and measuring the seedlings with mutant phenotype?

Results: If I remember correctly, the hmr mutant does not have a dark phenotype at normal growth temperature and I would expect that also at 27 °C there is no difference compared to the wild type in the dark ... can the authors confirm this?

Results: "To test this possibility, we treated wild-type Col-0 seedlings with lincomycin..."  the authors compare lincomycin treated Col-0 to non-treated hmr mutants. I think for consistency they should compare lincomycin treated Col-0 with lincomycin treated hmr.

Results: "In contrast, the steady-state protein level of PIF4 dramatically decreased in *hmr-5* under Rc, LD, and SD conditions."  Compared to what did the level of PIF4 decrease? Compared to the wild-type?  please reword to make this clear. Same in the next sentence "... remained the same in..."; the same as in what?

Results: "These results are consistent with the more pronounced phenotypes of *hmr-22* in Rc and LD conditions"  so the authors argue that PIF4 levels are related to the phenotype ... but then, why do *hmr-5* and *hmr-22* have a very similar phenotype which is clearly different from the wild type, while the PIF4 levels in *hmr-22* are much more similar to the wild type than to *hmr-5*? In addition, is there a difference between PIF4 protein levels in dark-grown *hmr-5*, *hmr-22*, and wild type?

Results: Figure 3c and 3d: Fig. 3c shows that PIF4 levels in *hmr-22* reach about 60% of the levels in wild type in Rc, while in Fig. 3d PIF4 is not detectable any more after 24 h at 27 °C. Does that mean that PIF4 levels are first upregulated (4 h), then downregulated (24 h), and then upregulated again?

Discussion: "PHYB's thermosensory role during daytime has been poorly recognized, largely because of the minimal temperature response in hypocotyl elongation under LD conditions in the white light, and the notion that PHYB's temperature-dependent dark-reversion rate operates only in the dark by influencing how fast the active PHYB diminishes at night."  I do not agree with the last statement that phyB's temperature-dependent dark-reversion only operates in the dark; Legris et al., 2016, Science, 354, 897–900 have shown that temperature also affects the Pfr/Ptot levels in light-grown seedlings. Furthermore, if the role of phyB in temperature sensing under LD conditions in white light, i.e. under "natural conditions" is minimal and only visible in red light (i.e. conditions under which plants normally do not grow), one may wonder whether temperature sensing by phyB in light is relevant at all given that plants do not grow in red light under natural conditions ... thus, the authors would argue that their own work is not relevant, which is not true, and therefore I suggest to reword this part.

Discussion: "The striking contrast in the hypocotyl thermoresponse between our results in R light vs. those from the studies in the white light supports the idea that the reduced hypocotyl thermoresponse observed in the previous studies was likely due to the white light conditions ...".  as mentioned in one of the previous comments, I think the authors have to include white light controls in the experiment in Fig. 1a-d to make this conclusion and to rule out that other factors than the light conditions different between their experiments and the previous studies are responsible for the result.

Discussion: "HMR mediates daytime thermosensing by ..." and "... PIF4-mediated thermosensing depends ..."  this is a bit misleading and one might have the impression that HMR and PIF4 are sensing the temperature, which the authors do not want to claim (as I understand). Thus, please make clear that HMR and PIF4 play a role in downstream signalling in phyB mediated thermosensing. Replacing "thermosensing" by "thermomorphogenesis" would also solve the problem.

Discussion: "We were not able to detect in vivo HMR-PIF4 interaction by immunoprecipitation, which might be due to low levels of HMR in the nucleus or the dynamic nature of the interaction."  this is a critical point: can the authors distinguish between HMR in the nucleus and in the plastids? If the authors are right and the fraction in the nucleus is much smaller than the fraction in the plastids, then the immunoblots in Fig. 5 would be problematic and they would not prove that levels of the different HMR versions in the nucleus (endogenous, HMR-HA, HMR22-HA-VP16) are similar (they would rather reflect the abundance of the major HMR fraction which would be the plastid fraction).

Discussion: "The *hmr* mutants have long hypocotyls in 21 °C mainly due to the accumulation of

PIF3..."  I don't think that reference 50 shows that the hypocotyl phenotype is mainly due to accumulation of PIF3; this conclusion would require an *hmr pif3* double mutant, which then should fully suppress the hypocotyl growth phenotype of *hmr*. Is there a reference showing this? Western blot data for PIF3 levels only show that there is correlation between *hmr* mutant and PIF3 levels but they cannot prove that the altered PIF3 levels are the (main) reason for the increased hypocotyl growth.

Discussion: "These results are consistent with our previous findings that the hypocotyl phenotype of *hmr* at 21 °C is caused by PIF1 and PIF3 accumulation in the nucleus..."  same comment as above: wouldn't that require double or even an *hmr pif1 pif3* triple mutant? Please reword or give a reference.

Discussion: "Given the critical function of HMR in thermomorphogenesis, this study provides genetic evidence supporting an important role of photobodies in thermosensing in the daytime."  present and previous work by the authors shows that HMR plays a role in formation of photobodies and thermomorphogenesis suggesting that there is a link between photobodies and thermomorphogenesis. However, I don't think that there is a strict prove showing that these events are causally related.

Discussion: "In conclusion, this study reveals a thermosensing role of PHYB in the daytime."  I think this has already been suggested by Legris et al., 2016, *Science*, 354, 897–900.

Discussion: "... a novel thermosensory mechanism in which HMR's TAD facilitates the activation of thermo-responsive PIF4 target-genes as well as PIF4 accumulation."  I would avoid "thermosensing mechanism"; HMR does not sense the temperature, it is a downstream signalling component in phyB mediated thermosensing.

Reviewer #2 (Remarks to the Author):

This manuscript addresses the interesting and timely question of how plants sense elevated temperature during the daytime. It builds on two recently published manuscripts in *Science*, showing that inactivation of the phyB photoreceptor by high temperature during the night (Jung et al. 2016) and in low levels of light (Legris et al. 2016) drive hypocotyl elongation via the transcription factor, PIF4. Experiments are performed at low (10 $\mu\text{molm}^{-2}\text{s}^{-1}$) red light to remove blue light-mediated suppression of the PIF4 activity. The authors demonstrate that PIF4 binds to the transcriptional activator HEMERA and the transcriptional activator domain (TAD) of HEMERA is required for the stabilization and activation of PIF4 at high temperature. The authors claim that phyB acts as a daytime sensor of high temperature and that high temperature mediated-inactivation of phyB drives hypocotyl elongation via HEMERA-controlled stabilization and activation of PIF4.

The manuscript has a number of strengths. It is clearly written and addresses a very timely question in plant biology. The manuscript provides solid evidence for the role of HEMERA in high-temperature mediated stabilisation and activation of PIF4. This is novel information which adds to the thermomorphogenesis field. Analyses appear appropriate and technical details appear to be sufficient for replication. There are however a number of issues to be addressed:

1. The novelty of the paper appears to centre on the role of phyB as a 'daytime' sensor of high temperature, to accompany the findings of Jung et al. (2016) that phyB conversion to the inactive Pr form is accelerated during the night. This conclusion is not novel. The authors use very low light levels in their experiments and their findings therefore support the published observations of Legris et al (2016). The use of red light also provides no broader context in which the importance of HMR in thermomorphogenesis can be assessed. Although the authors have justified their

conditions as designed to remove antagonism from cry1, clear thermomorphogenesis responses have been recorded in white light at much higher light levels in both long days and in continuous light. What is the phenotype of hmr mutants in these conditions?

2. The authors have already published that PIF4 interacts with HMR via the APB domain, using GST pull down assays (Qiu et al. 2015, Plant Cell). Figure 3 e is therefore not novel, as the findings replicate those of Figure 5 in their previously published manuscript. The authors do, however, show a role for this interaction in PIF4 stabilization in this study.

Additional minor issues:

1. The introduction mentions TOC 1 as a protein which physically binds to PIF4 during the early night to repress PIF4 activity. ELF3 should also be discussed in this context (Nieto et al 2015, Current Biology)

2. Harvest times for LD- and SD-grown seedlings should be specified in the legend for Figure 3a and 4a.

3. The Figure 3 legend defines abbreviations that are missing from the figure (PIR, GLU, NLS).

Response to Reviewers

Reviewer #1

The authors have previously shown that HMR plays a role in phyB downstream signalling. In the present manuscript they extend the role of HMR and show that it is involved in phyB mediated temperature-regulation of hypocotyl growth by controlling PIF4 accumulation. Overall, this is an interesting manuscript and I enjoyed reading it. Below I have a couple of comments and questions:

1. *Abstract: The authors point out that temperature sensing by phyB is thought to operate mainly at night. However, I think it is worth mentioning that a function of phyB in daytime temperature sensing has been suggested as well and that even in light, temperature affects the levels of active phyB (Legris et al., 2016, Science, 354, 897–900: "...Temperature affects the Pfr status of phyB mainly via kr1 in the light (Fig. 1) and via kr2 during the night (19).")*

Response: We have added a sentence in the Abstract: "PHYB should theoretically be able to sense temperature during the daytime".

2. *Abstract: "Fusing the VP16 TAD..."  only when reading the full manuscript it became clear to me that the VP16 TAD was fused to the HMR22 mutant protein  please reword so that this is clear from the abstract.*

Response: We have changed the sentence to "Fusing the TAD of VP16 to HMR22 rescued both its defects".

3. *Introduction: "It has been thought that PHYB plays a minor role in temperature-sensing during daytime 45,46"  I agree that this has been assumed but reference 19 shows that temperature affects the Pfr status of phyB in light; thus, there has been evidence that phyB also works as temperature sensor during daytime.*

Response: We have added a sentence at the beginning of the paragraph: "Because PHYB's dark-reversion rate can be influenced by temperature in the light¹, PHYB should theoretically be able to sense temperature during daytime in LD conditions."

4. *Results: "We hypothesized that the daytime thermosensory role of PHYB in LD conditions might be masked by the blue light photoreceptor CRY1 in the white light."  to test this hypothesis the authors performed experiments in red light, which is fine. However, an additional experiment would be to use the cry1 mutants in white light, which would further strengthen their conclusions.*

Response: We have performed an additional set of experiments with Col-0 and *cry1* in four different conditions in the white light. The results, now shown in Fig. 1., indicate that the warm temperature-induced hypocotyl response in the white light is quite complex, as it can be largely influenced by light intensity and developmental stage as well as an interplay between the growth conditions and CRY1 signaling. Under conditions where the warm-temperature response is

repressed, such as in 4-d-old seedlings grown in 100 $\mu\text{mol m}^{-2} \text{s}^{-1}$ and 40 $\mu\text{mol m}^{-2} \text{s}^{-1}$ white light, CRY1 played a predominant role in inhibiting the warm temperature-induced hypocotyl elongation response (Fig. 1). These new results, which are also consistent with the published results by Ma et al. (2016)², confirm our hypothesis and advocate the usage of monochromatic red light conditions for discerning PHYB's role in thermosensing in the light.

5. *Results: "... under SD, LD, and continuous light conditions with monochromatic R light."  from this it is not clear whether they also used red light for the LD and SD treatments. Based on the figure legend I think they also used red light for SD and LD  please reword in the result section so that this is clear.*

Response: We have revised the sentence to "we examined the hypocotyl responses of Col-0 in 21°C and 27°C under SD, LD, and continuous light conditions with the light periods in monochromatic R light."

6. *Results: "These results support our hypothesis that the daytime hypocotyl response is masked in the white light."  If I understand correctly, the authors performed all their experiments in cR or in LD/SD with R/D cycles. To make the comparison to white light, I think it is essential to include the white light control in their experiments, i.e. comparing their own red light data to white light data in the literature is problematic since differences between their own experimental setup and the setup in the literature not related to red/white light might affect seedling growth.*

Response: The reviewer is correct. As we show in Fig. 1., the temperature response was largely dependent on experimental conditions. Nonetheless, the new results show that both PHYB and HMR play important roles in photomorphogenesis in the white light (Fig. 2e).

7. *Results: "... we examined the hypocotyl response of the null hmr-5 mutant in continuous light Rc"  I thought that hmr null mutants are seedling lethal ... if this is true, how could they measure hypocotyl growth? Using a segregating population, sorting out seedlings with a WT phenotype, and measuring the seedlings with mutant phenotype?*

Response: The reviewer is correct. We maintain null alleles of *hmr* in heterozygous populations and measure the homozygous albino seedlings in a segregating population.

8. *Results: If I remember correctly, the hmr mutant does not have a dark phenotype at normal growth temperature and I would expect that also at 27 °C there is no difference compared to the wild type in the dark ... can the authors confirm this?*

Response: Yes, the *hmr* mutants do not have a hypocotyl phenotype in the dark under both 21°C and 27°C.

9. *Results: "To test this possibility, we treated wild-type Col-0 seedlings with lincomycin..."  the authors compare lincomycin treated Col-0 to non-treated hmr mutants. I think for consistency they should compare lincomycin treated Col-0 with lincomycin treated hmr.*

Response: Lincomycin treatment blocks plastid translation. Because the core subunits of the plastid-encoded plastid RNA polymerase (PEP) are encoded by the plastome and translated in plastids, lincomycin treatment also blocks the expression of the PEP, which initiates the plastid-to-nucleus retrograde GUN signaling³. The lincomycin experiment was to test whether albino mutants with defects in the PEP have a normal hypocotyl response to warm temperatures. The answer is yes (Fig. 2). This result supports the idea that the temperature phenotype of the *hmr-22* mutant, which is also defective in the PEP function, is unlikely the result of its chloroplast defects. Treating the *hmr-22* mutant with lincomycin would be redundant because the PEP is already unfunctional in *hmr-22*. But, to answer the reviewer question, we went ahead and did the experiment anyway. As shown in the following figure, the Relative Response of *hmr-22* to warm temperatures in the presence of lincomycin was 40%, which is quite similar to the 32% Relative Response without lincomycin. These results show that the temperature phenotype of *hmr-22*, as expected, is not dependent on lincomycin treatment.

Fig. Hypo-sensitivity of *hmr-22* to warm temperature is not affected by the lincomycin treatment. Hypocotyl length of 4-d-old Col-0, Col-0 treated with 220 $\mu\text{g/ml}$ lincomycin, *hmr-22*, and *hmr-22* treated with 220 $\mu\text{g/ml}$ lincomycin was measured and compared. Seedlings were grown in 10 $\mu\text{mol m}^{-2} \text{s}^{-1}$ Rc at 21°C and 27°C. The Relative Response (magenta bars) was defined as relative hypocotyl response to 27°C of each line compared with that of Col-0 without lincomycin treatment. Error bars for the hypocotyl length represent SD ($n > 30$); error bars for the relative responses represent SD of three biological replicates. Black numbers show the mean \pm SD values of the absolute responses of Col-0 ($n = 3$). Purple numbers show the mean \pm SD values of relative responses and different letters denote statistically significant differences in Relative Responses (ANOVA, Tukey's HSD, $p < 0.01$, $n = 3$).

10. Results: "In contrast, the steady-state protein level of PIF4 dramatically decreased in *hmr-5* under Rc, LD, and SD conditions."  Compared to what did the level of PIF4 decrease? Compared to the wild-type?  please reword to make this clear. Same in the next sentence "... remained the same in..."; the same as in what?

Response: We have revised the two sentences by adding "compared with that of Col-0..."

11. Results: "These results are consistent with the more pronounced phenotypes of *hmr-22* in Rc and LD conditions"  so the authors argue that PIF4 levels are related to the phenotype ... but then, why do *hmr-5* and *hmr-22* have a very similar phenotype which is clearly different from the wild type, while the PIF4 levels in *hmr-22* are much more similar to the wild type than to *hmr-5*? In addition, is there a difference between PIF4 protein levels in dark-grown *hmr-5*, *hmr-22*, and wild type?

Response: Thermomorphogenesis in *Arabidopsis* is centrally regulated by PIF4⁴⁻⁶. Collaborating with the previous studies, our results also show a correlation between the level of PIF4 and the warm temperature responses in *hmr-5* and *hmr-22*. The warm-temperature-dependent hypocotyl-elongation responses of *hmr-5* and *hmr-22* are quite different (Fig. 2). While the null allele *hmr-5* retained only a 8% Relative Response, *hmr-22* had a 28% Relative Response in Rc (Fig. 2b). Therefore, the hypocotyl phenotype of *hmr-22* is much weaker than *hmr-5*, which is consistent with more PIF4 in *hmr-22* (1.5) than in *hmr-5* (0.1) in Rc (Fig. 4c). Moreover, the level of PIF4 also correlates with the warm temperature response by comparing *hmr-22* in different photoperiods. For example, in SD, the level of PIF4 in *hmr-22* (1.8) was similar to that of Col-0 (2), and *hmr-22* maintained a 62% Relative Response compared with Col-0 (Fig. 2d). In contrast, in LD, the level of PIF4 declined more in *hmr-22* (1.5) compared with Col-0 (2.2) (Fig. 4c), and *hmr-22* also had a more reduced Relative Response at 41% (Fig. 2d). The levels of PIF4 also correlated with the expression of its target genes (Fig. 5). We have not looked at PIF4 levels in the dark because the *hmr* mutants do not have a phenotype in the dark⁷. In addition, here we wanted to focus on the mechanism of temperature sensing in the light or the daytime.

12. Results: Figure 3c and 3d: Fig. 3c shows that PIF4 levels in *hmr-22* reach about 60% of the levels in wild type in Rc, while in Fig. 3d PIF4 is not detectable any more after 24 h at 27 °C. Does that mean that PIF4 levels are first upregulated (4 h), then downregulated (24 h), and then upregulated again?

Response: The reviewer brought up an interesting point. We think the two experimental conditions were quite different: one was to measure the steady-state level of PIF4 at a warm temperature (new Fig. 4c) and the other the dynamics of PIF4 in response to warm temperature (new Fig. 4d). It might mean that it takes more than 24 h for PIF4 to reach its equilibrium level at 27°C.

13. Discussion: "PHYB's thermosensory role during daytime has been poorly recognized, largely because of the minimal temperature response in hypocotyl elongation under LD conditions in the white light, and the notion that PHYB's temperature-dependent dark-reversion rate operates only in the dark by

influencing how fast the active PHYB diminishes at night."  I do not agree with the last statement that phyB's temperature-dependent dark-reversion only operates in the dark; Legris et al., 2016, Science, 354, 897–900 have shown that temperature also affects the Pfr/Ptot levels in light-grown seedlings. Furthermore, if the role of phyB in temperature sensing under LD conditions in white light, i.e. under "natural conditions" is minimal and only visible in red light (i.e. conditions under which plants normally do not grow), one may wonder whether temperature sensing by phyB in light is relevant at all given that plants do not grow in red light under natural conditions ... thus, the authors would argue that their own work is not relevant, which is not true, and therefore I suggest to reword this part.

Response: With the new data in the white light (Fig. 1), this paragraph has been substantially revised.

14. Discussion: *"The striking contrast in the hypocotyl thermoresponse between our results in R light vs. those from the studies in the white light supports the idea that the reduced hypocotyl thermoresponse observed in the previous studies was likely due to the white light conditions ...".  as mentioned in one of the previous comments, I think the authors have to include white light controls in the experiment in Fig. 1a-d to make this conclusion and to rule out that other factors than the light conditions different between their experiments and the previous studies are responsible for the result.*

Response: Please see the responses to Question 4 and 6.

15. Discussion: *"HMR mediates daytime thermosensing by ..." and "... PIF4-mediated thermosensing depends ..."  this is a bit misleading and one might have the impression that HMR and PIF4 are sensing the temperature, which the authors do not want to claim (as I understand). Thus, please make clear that HMR and PIF4 play a role in downstream signalling in phyB mediated thermosensing. Replacing "thermosensing" by "thermomorphogenesis" would also solve the problem.*

Response: We have made the changes based on the reviewer comment.

16. Discussion: *"We were not able to detect in vivo HMR-PIF4 interaction by immunoprecipitation, which might be due to low levels of HMR in the nucleus or the dynamic nature of the interaction."  this is a critical point: can the authors distinguish between HMR in the nucleus and in the plastids? If the authors are right and the fraction in the nucleus is much smaller than the fraction in the plastids, then the immunoblots in Fig. 5 would be problematic and they would not prove that levels of the different HMR versions in the nucleus (endogenous, HMR-HA, HMR22-HA-VP16) are similar (they would rather reflect the abundance of the major HMR fraction which would be the plastid fraction).*

Response: We have shown previously that HMR could not be overexpressed - the levels of recombinant HMR in transgenic lines, such as *HMR-HA/hmr-5*, are always similar to that of endogenous HMR in Col-0⁸. Our unpublished data show that this is because lines accumulating excess amount of HMR are male-sterile. HMR is dual-targeted to the nucleus and plastids, and nuclear and plastidial HMR proteins have the same molecular mass⁹⁻¹². We have not yet found a condition where the nuclear-and-plastidial partitioning of HMR is altered. Our previous studies

have shown that the function of nuclear HMR in the regulation of hypocotyl elongation can be separated from the role of plastidial HMR as an essential component of the plastid RNA polymerase, because the nuclear function is PIF-dependent⁷. Here we showed that the warm temperature-dependent hypocotyl response is not dependent on functional chloroplasts (Fig. 2), suggesting that the thermomorphogenesis phenotype of *hmr-22* is due to defects of HMR's nuclear function. We have tried extensively to detect in vivo HMR-PIF4 interaction by immunoprecipitation. However, these experiments were unsuccessful. One possible reason is due to the minimal amount of HMR in the nucleus. Another possibility is that the HMR-PIF4 interaction might not be stable enough to be captured by immunoprecipitation. We then tried the alternative approach to see whether fusing a functional transactivation domain to HMR22 could rescue its defect in the activation of the temperature-responsive PIF4 direct target genes. The result that HMR22-HA-VP16 rescues *hmr-22*'s defects in the activation of the PIF4 target-genes support the model that HMR participates in PIF4-mediated activation of temperature-responsive genes in vivo. It was surprising that HMR22-HA-VP16 can also rescue the PIF4 accumulation defect of *hmr-22*, because it could not rescue the defect of PIF3 degradation⁷. These results indicate distinct roles of HMR in the regulation of the stability of PIF3 and PIF4.

17. Discussion: "The *hmr* mutants have long hypocotyls in 21 °C mainly due to the accumulation of PIF3..."  I don't think that reference 50 shows that the hypocotyl phenotype is mainly due to accumulation of PIF3; this conclusion would require an *hmr pif3* double mutant, which then should fully suppress the hypocotyl growth phenotype of *hmr*. Is there a reference showing this? Western blot data for PIF3 levels only show that there is correlation between *hmr* mutant and PIF3 levels but they cannot prove that the altered PIF3 levels are the (main) reason for the increased hypocotyl growth.

Response: We have reworded this part to "These distinct functions of HMR on the stability of PIF3 and PIF4 provide an explanation for the contrasting phenotypes of the *hmr* mutants under 21°C and 27°C: The long-hypocotyl phenotype of *hmr* mutants in 21°C could mainly due to the accumulation of PIF3⁷, whereas the short-hypocotyl phenotype in 27°C is caused by the defect in PIF4 accumulation (Fig. 3a)".

18. Discussion: "These results are consistent with our previous findings that the hypocotyl phenotype of *hmr* at 21 °C is caused by PIF1 and PIF3 accumulation in the nucleus..."  same comment as above: wouldn't that require double or even an *hmr pif1 pif3* triple mutant? Please reword or give a reference.

Response: Hypocotyl growth is promoted by all four PIFs. We have shown previously that the *hmr* mutants accumulate PIF1 and PIF3 at 21°C in the light^{7,9}. We did not look at the level of PIF4, because PIF4 antibody was not available until recently. We had thought that PIF4 would behave similarly as PIF1 and PIF3 and also accumulate in the *hmr* mutants. Surprisingly, here we show that PIF4 cannot accumulate in the *hmr* mutants (Fig. 4c,d). Therefore, it is conceivable

that the long-hypocotyl phenotypes of the *hmr* mutants at 21°C must not be due to PIF4 and therefore should mainly be due to the accumulation of PIF1 and PIF3.

19. Discussion: "Given the critical function of HMR in thermomorphogenesis, this study provides genetic evidence supporting an important role of photobodies in thermosensing in the daytime."  present and previous work by the authors shows that HMR plays a role in formation of photobodies and thermomorphogenesis suggesting that there is a link between photobodies and thermomorphogenesis. However, I don't think that there is a strict proof showing that these events are causally related.

Response: We have changed the sentence to "Given the critical function of HMR in thermomorphogenesis, this study provides genetic evidence supporting a role of photobodies in thermosensing in the daytime, which is consistent with the dynamic changes of photobody morphology under different temperatures¹."

20. Discussion: "In conclusion, this study reveals a thermosensing role of PHYB in the daytime."  I think this has already been suggested by Legris et al., 2016, Science, 354, 897–900.

Response: We have changed the sentence to "this study demonstrates a thermosensing role of PHYB in the daytime".

21. Discussion: "... a novel thermosensory mechanism in which HMR's TAD facilitates the activation of thermo-responsive PIF4 target-genes as well as PIF4 accumulation."  I would avoid "thermosensing mechanism"; HMR does not sense the temperature, it is a downstream signalling component in phyB mediated thermosensing.

Response: We have changed it to "Our results support a novel PHYB-mediated temperature signaling mechanism in which..."

Reviewer #2:

This manuscript addresses the interesting and timely question of how plants sense elevated temperature during the daytime. It builds on two recently published manuscripts in Science, showing that inactivation of the phyB photoreceptor by high temperature during the night (Jung et al. 2016) and in low levels of light (Legris et al. 2016) drive hypocotyl elongation via the transcription factor, PIF4. Experiments are performed at low (10 μmolm⁻²s⁻¹) red light to remove blue light-mediated suppression of the PIF4 activity. The authors demonstrate that PIF4 binds to the transcriptional activator HEMERA and the transcriptional activator domain (TAD) of HEMERA is required for the stabilization and activation of PIF4 at high temperature. The authors claim that phyB acts as a daytime sensor of high temperature and that high temperature mediated-inactivation of phyB drives hypocotyl elongation via HEMERA-controlled stabilization and activation of PIF4.

The manuscript has a number of strengths. It is clearly written and addresses a very timely question in plant biology. The manuscript provides solid evidence for the role of HEMERA in high-temperature mediated stabilisation and activation of PIF4. This is novel information which adds to the thermomorphogenesis field. Analyses appear appropriate and technical details appear to be sufficient for replication. There are however a number of issues to be addressed:

- 1. The novelty of the paper appears to centre on the role of phyB as a 'daytime' sensor of high temperature, to accompany the findings of Jung et al. (2016) that phyB conversion to the inactive Pr form is accelerated during the night. This conclusion is not novel. The authors use very low light levels in their experiments and their findings therefore support the published observations of Legris et al (2016). The use of red light also provides no broader context in which the importance of HMR in thermomorphogenesis can be assessed. Although the authors have justified their conditions as designed to remove antagonism from cry1, clear thermomorphogenesis responses have been recorded in white light at much higher light levels in both long days and in continuous light. What is the phenotype of hmr mutants in these conditions?*

Response: We have performed new experiments in different conditions under the white light (Fig. 1 and Fig. 2e). The results of these experiments showed that the temperature response in the white light is largely influenced by light intensity and developmental stage as well as the interplay between the growth conditions and CRY1 signaling. CRY1 signaling could inhibit the temperature response in certain conditions but promote it in others, suggesting complex interactions between PHYB and CRY1 signaling that are dependent on light intensity and seedling age. These results also showed that PHYB and HMR are required for thermomorphogenesis under LD conditions in the white light (Fig. 2e). Moreover, given the complex effects of CRY1 signaling on the thermoresponses in the white light, these new data, as well as our mutant analysis in R light, support the advantage of using monochromatic R light to discern PHYB's function in thermosensing in long-day and continuous light conditions.

- 2. The authors have already published that PIF4 interacts with HMR via the APB domain, using GST pull down assays (Qiu et al. 2015, Plant Cell). Figure 3 e is therefore not novel, as the findings replicate those of Figure 5 in their previously published manuscript. The authors do, however, show a role for this interaction in PIF4 stabilization in this study.*

Response: Our previous study⁷ showed only that PIF4's APB could interact with HMR. However, it was still unclear whether APB was the only interacting domain. Here we further characterized the interaction between HMR and a series of N- and C-terminal truncated fragments of PIF4, these results demonstrate that the HMR-PIF4 interaction is mainly mediated by PIF4's APB motif. This study also reveals the biological significance of the HMR-PIF4 interaction in thermomorphogenesis.

- 3. The introduction mentions TOC 1 as a protein which physically binds to PIF4 during the early night to repress PIF4 activity. ELF3 should also be discussed in this context (Nieto et al 2015, Current Biology)*

Response: We have added this reference.

4. Harvest times for LD- and SD-grown seedlings should be specified in the legend for Figure 3a and 4a.

Response: We have added the sample collection time in the legends for the new Fig. 4a and 5a.

5. The Figure 3 legend defines abbreviations that are missing from the figure (PIR, GLU, NLS)

Response: We have removed these extra abbreviations.

References:

1. Legris, M. *et al.* Phytochrome B integrates light and temperature signals in Arabidopsis. *Science* **354**, 897–900 (2016).
2. Ma, D. *et al.* Cryptochrome 1 interacts with PIF4 to regulate high temperature-mediated hypocotyl elongation in response to blue light. *Proceedings of the National Academy of Sciences* **113**, 224–229 (2016).
3. Woodson, J. D., Perez-Ruiz, J. M., Schmitz, R. J., Ecker, J. R. & Chory, J. Sigma factor-mediated plastid retrograde signals control nuclear gene expression. *Plant J.* **73**, 1–13 (2012).
4. Gangappa, S. N. & Kumar, S. V. DET1 and HY5 Control PIF4-Mediated Thermosensory Elongation Growth through Distinct Mechanisms. *Cell Rep.* **18**, 344–351 (2017).
5. Kumar, S. V. *et al.* Transcription factor PIF4 controls the thermosensory activation of flowering. *Nature* **484**, 242–245 (2012).
6. Koini, M. A. *et al.* High temperature-mediated adaptations in plant architecture require the

- bHLH transcription factor PIF4. *Curr. Biol.* **19**, 408–413 (2009).
7. Qiu, Y. *et al.* HEMERA Couples the Proteolysis and Transcriptional Activity of PHYTOCHROME INTERACTING FACTORS in Arabidopsis Photomorphogenesis. *Plant Cell* **27**, 1409–1427 (2015).
 8. Galvao, R. M. *et al.* Photoactivated phytochromes interact with HEMERA and promote its accumulation to establish photomorphogenesis in Arabidopsis. *Genes Dev.* **26**, 1851–1863 (2012).
 9. Chen, M. *et al.* Arabidopsis HEMERA/pTAC12 initiates photomorphogenesis by phytochromes. *Cell* **141**, 1230–1240 (2010).
 10. Nevarez, P. A. *et al.* Mechanism of Dual Targeting of the Phytochrome Signaling Component HEMERA/pTAC12 to Plastids and the Nucleus. *Plant Physiol.* **173**, 1953–1966 (2017).
 11. Pfalz, J. *et al.* ZmpTAC12 binds single-stranded nucleic acids and is essential for accumulation of the plastid-encoded polymerase complex in maize. *New Phytol.* **206**, 1024–1037 (2015).
 12. Pfalz, J., Liere, K., Kandlbinder, A., Dietz, K. J. & Oelmüller, R. pTAC2, -6, and -12 are components of the transcriptionally active plastid chromosome that are required for plastid gene expression. *Plant Cell* **18**, 176–197 (2006).

Reviewers' comments:

Reviewer #1 (Remarks to the Author):

The authors addressed all my comments and I have only two comments on the revised version of the manuscript:

- The authors use the *hmr-5* allele, which is lethal. As they describe in the rebuttal letter, they maintain the allele in segregating populations and only use the homozygous seedlings for the experiments. If not already done, please include this information in the Method section.

- In Fig. 2e I do not understand the % values; for instance, for *phyB* the hypocotyl length at 21 °C is less than 5 mm and at 27 °C it is at least 9 mm, i.e. the increase is >80% ... how does that fit to 33%? Similar for *phyA/phyB* where hypocotyl length is about 4.5 mm vs. 6.5 ... which would be an increase of more 40% and not only 19%. Please clarify this issue.

Reviewer #2 (Remarks to the Author):

1. The authors have addressed a number of my concerns. However, one significant issue I raised in my review was the key conclusion that 'PHYB controls daytime temperature-dependent hypocotyl elongation in LD conditions mainly by regulating the activity and stability of HMR' (page 15). As this conclusion refers to natural (ie. white light) conditions, it is necessary to show the phenotype of *hmr* mutants in long day photoperiods of white light. The authors appear to have shown mutant phenotypes in continuous white light (Figure 2e- graph title and legend), yet the response to reviewers' letter states that this experiment was performed in long days- 'These results showed that PHYB and HMR are required for thermomorphogenesis under LD conditions'. Please can this be clarified and LD data shown or labelling/legend corrected. A similar issue arises with Figure 1. The graph title and figure legend state continuous light, yet the discussion describes the experiment as having been performed in long days- 'We show here that this is mainly due to the complex factors contributing to the temperature responses in LD conditions in the white light, particularly the interplay between growth conditions and *cry1* signaling (Fig1)'. Col-0 data for Figure 1 condition (D) are replicated in Figure 2E, suggesting that these two experiments were performed together.

2. In agreement with reviewer 1, it could be made clearer that experiments in figures 2c and 2d were performed in LD and SD cycles of red light. Specifying 'R' on the images and graphs for figure 2c,d would aid clarity for the reader.

3. In figure 2e, a significant high temperature response is observed in the *phyB* mutant. The authors should highlight the significance of this finding- that high temperature-mediated elongation growth in white light must include *phyB*-independent mechanism(s). This does reduce the importance of their findings but provides a more balanced interpretation and overall view of thermosensing.

Response to Reviewers

We would like to thank both reviewers for their thorough reviews of the manuscript and for the many constructive comments and suggestions that have helped us to strengthen the conclusion and improve the clarity of the manuscript.

Reviewer #1

The authors addressed all my comments and I have only two comments on the revised version of the manuscript:

*-The authors use the *hmr-5* allele, which is lethal. As they describe in the rebuttal letter, they maintain the allele in segregating populations and only use the homozygous seedlings for the experiments. If not already done, please include this information in the Method section.*

Response: We have added a sentence in the Methods: “Because *hmr-5* is albino and seedling lethal, homozygous *hmr-5* seedlings used in this study were from segregating populations.”

*- In Fig. 2e I do not understand the % values; for instance, for *phyB* the hypocotyl length at 21 °C is less than 5 mm and at 27 °C it is at least 9 mm, i.e. the increase is >80% ... how does that fit to 33%? Similar for *phyA/phyB* where hypocotyl length is about 4.5 mm vs. 6.5 ... which would be an increase of more 40% and not only 19%. Please clarify this issue.*

Response: The warm temperature-dependent hypocotyl elongation response of a particular genotype has usually been characterized as the percentage of increase in hypocotyl length in 27°C vs. 21°C. However, this percentage value does not reflect how the response compares with that in the control line or the wild-type, and therefore is not intuitive enough to understand the mutant phenotype. To circumvent this problem, we introduced a new parameter, termed Relative Response, to represent how the temperature response in mutants is relative to that in the wild-type. A Relative Response of a mutant is calculated by dividing the percentage of hypocotyl response of the mutant by that of the wild-type. For example, in Fig. 2e, the percentages of hypocotyl increase for *phyB* and Col-0 were 90% and 273%, and thus the Relative Response of *phyB* is $90/273=33\%$, which indicates that under the continuous white light condition the *phyB* mutant showed only 33% of the temperature response in Col-0.

Reviewer #2:

*1. The authors have addressed a number of my concerns. However, one significant issue I raised in my review was the key conclusion that ‘PHYB controls daytime temperature-dependent hypocotyl elongation in LD conditions mainly by regulating the activity and stability of HMR’ (page 15). As this conclusion refers to natural (ie. white light) conditions, it is necessary to show the phenotype of *hmr* mutants in long day photoperiods of white light. The authors appear to have shown mutant phenotypes in continuous white light (Figure 2e- graph title and legend), yet the response to reviewers’ letter states that this experiment was performed in long days- ‘These results showed that PHYB and HMR are required for thermomorphogenesis under LD conditions’. Please can this be clarified and LD data shown or*

labelling/legend corrected. A similar issue arises with Figure 1. The graph title and figure legend state continuous light, yet the discussion describes the experiment as having been performed in long days- 'We show here that this is mainly due to the complex factors contributing to the temperature responses in LD conditions in the white light, particularly the interplay between growth conditions and cry1 signaling (Fig1)'. Col-0 data for Figure 1 condition (D) are replicated in Figure 2E, suggesting that these two experiments were performed together.

Response: The main contribution of this study is showing that PHYB and HMR (a necessary component mediating thermomorphogenesis) play critical roles in thermo-responses during the daytime - in the light. This is a significant departure from the current view that PHYB senses temperature mainly during the nighttime - in the dark. Daytime and nighttime temperature-sensing by PHYB represent two distinct mechanisms for the regulation of plant growth and occur under different photoperiods. Nighttime hypocotyl elongation mainly occurs in SD conditions at the end of night when PHYB has become inactive; whereas daytime hypocotyl elongation primarily occurs in LD (continuous light is a special LD condition) during the daytime when PHYB remains active. We demonstrate here that, under continuous white light, continuous red light, and LD in R red light conditions, Arabidopsis seedlings can sense temperature changes through PHYB. Our data show that the warm temperature response is more pronounced under continuous R light conditions (Fig. 2). That's the reason why I chose to use continuous R light to demonstrate that PHYB can sense temperature changes in the light. Our data also show complex effects by CRY1 signaling in white light conditions (Fig. 1), which support the usage of monochromatic light to dissect the specific effects by PHYB. In our opinion, dissecting the specific effects of PHYB and CRY1 under monochromatic light conditions is a prerequisite for the understanding of their combined effects under natural light conditions. I agree that future investigations are warranted to unravel the complex relationship between PHYB and CRY1 in the temperature responses in white light conditions.

We thank this reviewer for these comments. We have revised the referred sentences to improve their clarity. (1) We have changed the sentence in the Discussion to “PHYB controls daytime temperature-dependent hypocotyl elongation in the light mainly by regulating the activity and stability of PIF4 through HMR”. (2) The legend of Figure 2 states for Figure 2e that “Seedlings were grown in $100 \mu\text{mol m}^{-2} \text{s}^{-1}$ continuous white light...” (3) We have changed the other sentence in the Discussion to “We show here that this is mainly due to the complex factors contributing to the temperature responses in the white light...”

The reviewer is correct that the experiments for Figure 1d and 2e were performed together, so the Col-0 control data are the same.

2. In agreement with reviewer 1, it could be made clearer that experiments in figures 2c and 2d were performed in LD and SD cycles of red light. Specifying 'R' on the images and graphs for figure 2c,d would aid clarity for the reader.

Response: We have made changes in the legend of figure 2: “c. ... seedlings grown in LD and SD with $10 \mu\text{mol m}^{-2} \text{s}^{-1}$ R light, labelled as R-LD and R-SD, respectively ...”. We have also specified R-LD and R-SD in panel 2c and 2d.

3. In figure 2e, a significant high temperature response is observed in the phyB mutant. The authors should highlight the significance of this finding- that high temperature-mediated elongation growth in white light must include phyB-independent mechanism(s). This does reduce the importance of their findings but provides a more balanced interpretation and overall view of thermosensing.

Response: We have added a sentence in the Results: “*phyB-9* in the white light showed a greater warm temperature response than in Rc (Fig. 2b), suggesting that the warm-temperature response in the white light is also mediated by sensors besides PHYB.”

We also added the following sentence in the Discussion: “Our results also show that PHYB plays a major role in continuous white light (Fig. 2e), although it is worth noting that under our white light condition *phyB-9* retained 33% of Col-0’s warm temperature response, suggesting that temperature sensing in the white light must also be mediated by other sensors besides PHYB.”